# Expressive probabilistic sampling in recurrent neural networks

**Shirui Chen**[*]
Department of Applied Mathematics
University of Washington, Seattle
`sc256@uw.edu`

**Linxing Preston Jiang**
Paul G. Allen School of Computer Science
& Engineering
University of Washington, Seattle
`prestonj@cs.washington.edu`

**Rajesh P. N. Rao**
Paul G. Allen School of Computer Science
& Engineering and Center for Neurotechnology
University of Washington, Seattle
`rao@cs.washington.edu`

**Eric Shea-Brown**
Department of Applied Mathematics
Computational Neuroscience Center
University of Washington, Seattle
`etsb@uw.edu`

## Abstract

In sampling-based Bayesian models of brain function, neural activities are assumed to be samples from probability distributions that the brain uses for probabilistic computation. However, a comprehensive understanding of how mechanistic models of neural dynamics can sample from arbitrary distributions is still lacking. We use tools from functional analysis and stochastic differential equations to explore the minimum architectural requirements for *recurrent* neural circuits to sample from complex distributions. We first consider the traditional sampling model consisting of a network of neurons whose outputs directly represent the samples (*sampler-only* network). We argue that synaptic current and firing-rate dynamics in the traditional model have limited capacity to sample from a complex probability distribution. We show that the firing rate dynamics of a recurrent neural circuit with a separate set of output units can sample from an arbitrary probability distribution. We call such circuits *reservoir-sampler networks* (RSNs). We propose an efficient training procedure based on denoising score matching that finds recurrent and output weights such that the RSN implements Langevin sampling. We empirically demonstrate our model's ability to sample from several complex data distributions using the proposed neural dynamics and discuss its applicability to developing the next generation of sampling-based Bayesian brain models.

## 1 Introduction

There is growing evidence that humans and other animals make decisions by representing uncertainty internally and carrying out probabilistic computations that approximate Bayesian inference [42, 30, 18, 23, 34]. How networks of neurons in the brain represent probability distributions for Bayesian inference has remained a major open question. There exist two major theories: one assumes that the neural activities encode the parameters of the underlying posterior distributions over sensory stimuli [5, 35, 52]. The other is the sampling-based hypothesis, which assumes that the neural responses can be interpreted as samples from a posterior distribution [28]. Under this hypothesis, recurrent neural circuits make use of their inherent stochasticity to produce samples from posterior distributions. The

---

[*]Corresponding author

37th Conference on Neural Information Processing Systems (NeurIPS 2023).

sampling-based theory has explained various experimental observations regarding neural variability [19, 20, 40], perceptual decision-making [24] and spontaneous cortical activity [7].

Many studies have proposed biologically plausible spiking rules and membrane dynamics models to implement sampling-based probabilistic inference. However, most of these studies mainly consider sampling from discrete Boltzmann distributions [12] and multivariate Gaussian distributions [17, 1, 38, 25], only match the first two moments of the distribution [19], or employ a Monte-Carlo approximation [29]. Although these studies use algorithmic substrates that can sample from any distribution with a density function *in theory*, it is not clear whether the underlying neural dynamics are capable of implementing a sufficiently expressive version of these sampling methods. Furthermore, natural image statistics are strongly non-Gaussian [39], and experimental evidence shows that humans use non-Gaussian prior representations to support cognitive judgments [27, 22]. It is known that deep artificial neural networks can be used to generate samples from complex data distributions [48, 49] using a "U-net" [43] backbone. However, the neural circuits in the cortex are highly coupled with an abundance of recurrent synapses. Therefore, an outstanding question for probabilistic computation in the brain is: what kind of recurrent neural network is capable of efficiently learning to produce samples from an arbitrarily complex posterior distribution?

In this paper, we study this question under the basic assumption that the dynamics of recurrent neural circuits can be described by stochastic differential equations (SDEs). Note that this assumption is common to a broad range of past research that uses rate-based neural dynamics to implement sampling-based coding[1, 19, 17]. Moreover, spike-based models [46, 45, 38] that implement balanced spiking networks (BSNs) [11, 13] essentially train spiking networks to follow underlying continuous-time SDEs, so our work applies to this line of research as well (details in Appendix G).

The contributions of this paper are as follows (Figure 1):

1. We establish the relationship between the sampling power of the neural dynamics and the ability of the dynamics to approximate the score function, which is the gradient of the log probability density function. (Section 3.1)

2. We show that the synaptic current dynamics of a network of neurons whose outputs directly represent the samples (traditional sampler-only network) is only able to approximate score functions that are in a finite-dimensional function space. (Section 3.2, Proposition 2)

3. We prove that the firing rate dynamics of our proposed reservoir-sampler network can sample from a distribution whose score function can approximate that of arbitrary target distributions (with mild restrictions) to arbitrary precision (Section 3.3, Theorem 3).

4. We derive a computationally efficient and biologically-plausible learning rule for our proposed model (Section 3.4) that sidesteps the demands of backpropagation through time, and we empirically demonstrate how our model can sample from several complex data distributions (Section 4).

And interpretation of our contributions in biological terms is as follows: 1. Flexible synaptic connectivity within a circuit itself is not enough to allow that circuit to flexibly produce arbitrary patterns of variability involving all of its neurons. 2. In order for circuit to achieve full flexibility in its output patterns, there need to be hidden variables involved, e.g. states of neurons in an upstream brain area or possibly non-synaptic signalling. If this condition is met, concrete and fairly efficient plasticity rules may be capable of shaping the output patterns as desired.

## 2    Background

### 2.1    Fokker-Planck equation and stationary distribution

We consider a general time-homogeneous SDE with drift vector $\mathbf{b}(X_t)$ and diffusion matrix $\Sigma = \frac{1}{2}\sigma\sigma^T$:

$$dX_t = \mathbf{b}(X_t)dt + \sigma dB_t \tag{1}$$

where $\sigma \in \mathbb{R}^{n \times m}$ is the diffusion coefficient, and $B_t$ is an $m$-dimensional standard Wiener process. The Fokker-Planck equation of this SDE describes the time evolution of the probability density $p(x, t)$ for a given SDE, assuming the initial density $p(x, 0)$ is known:

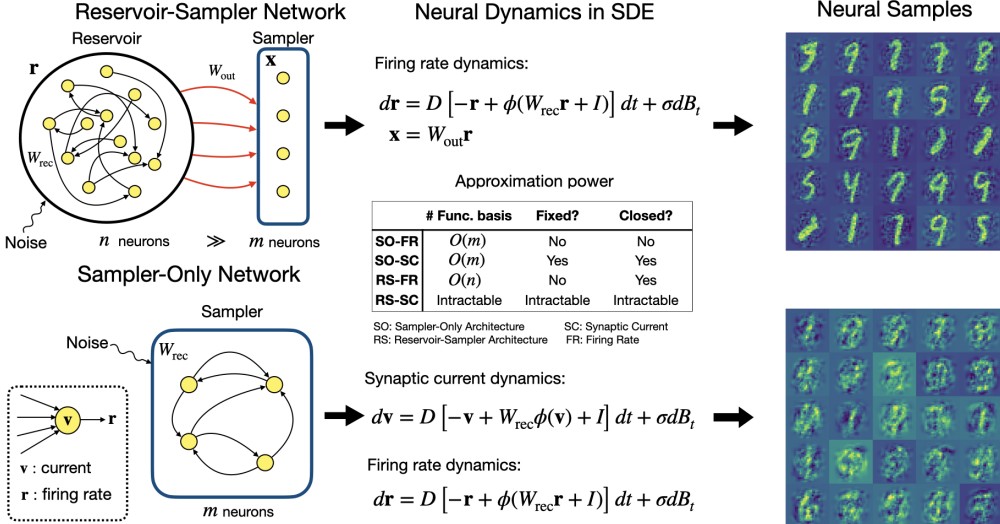

Figure 1: **Reservoir-Sampler Networks versus Traditional Sampler-Only Networks.** The sampler neurons which produce samples from the target distribution are present for both the sampler-only (SO) network and the reservoir-sampler (RS) network. The reservoir neurons (shown within the large black circle) are only present in the RS network. We explore the firing rate (FR) dynamics (Section 3.3) for both networks and the synaptic current (SC) dynamics (Section 3.2) only for the sampler-only network because the stationary distribution of the output neurons is intractable in the RS-SC case. We evaluate the approximation power of the function set represented by the drift term of the neural dynamics, and list the number of basis functions that span the set, whether these basis functions are fixed, and whether the function set is closed under addition (shown in the table).

$$\partial_t p = \nabla \cdot (\Sigma \nabla p - \mathbf{b}p) \tag{2}$$

where $\partial_t = \partial/\partial t$, $\nabla\cdot$ is the divergence operator, and $\nabla$ is the gradient operator. A stationary probability density function is one for which the right-hand-side of equation (2) is 0. Mild regularity conditions guaranteeing the existence and uniqueness of a stationary density function are discussed in, for example, Cerrai [14]. We assume these are satisfied by the SDEs considered in this paper. Moreover, to ensure ergodicity and a well-defined score function, we assume that $p$ is supported on $\mathbb{R}^n$, i.e. $p(x) > 0$ for all $x \in \mathbb{R}^n$.

As a special case, consider the following Langevin dynamics, in which the drift term is given by the gradient of the log stationary probability density $\nabla p(x)$,

$$dX_t = \nabla \log p(X_t)dt + \sqrt{2}dB_t. \tag{3}$$

It can be verified through the Fokker-Planck equation that $p(x)$ in the dynamics above is indeed a stationary probability density function of the dynamics. Therefore the Langevin dynamics can sample from the distribution $p(x)$ as $t \to \infty$. While the drift term in the Langevin dynamics is a gradient vector field, this is not true in general and is often not the case for recurrent neural dynamics (Proposition 5 in Appendix B).

## 2.2 Score-based generative modeling

If we would like a particular dynamics to implement Langevin dynamics, we need to fit the drift term of an SDE to the score $s_\theta(\mathbf{x}) = \nabla \log p(\mathbf{x})$ of the probability distribution that we are trying to sample from. This procedure of fitting the score function $s_\theta(\mathbf{x})$ is called score matching. In this section, we give a brief summary of one of the major methods of score matching that we will use in this paper, denoising score matching (DSM) [51]. The general idea of DSM is to match the score of a noise-perturbed version of the target distribution. More specifically, the explicit score matching loss

(left-hand side) and the denoising score matching loss (right-hand side) are related to each other as follows [51]:

$$\mathbb{E}_{q_\sigma(\tilde{\mathbf{x}})}\left[\frac{1}{2}\|s_\theta(\tilde{\mathbf{x}}) - \nabla_{\tilde{\mathbf{x}}}\log q_\sigma(\tilde{\mathbf{x}})\|^2\right] = \mathbb{E}_{q_\sigma(\tilde{\mathbf{x}},\mathbf{x})}\left[\frac{1}{2}\|s_\theta(\tilde{\mathbf{x}}) - \nabla_{\tilde{\mathbf{x}}}\log q_\sigma(\tilde{\mathbf{x}}|\mathbf{x})\|^2\right] + C_\sigma \quad (4)$$

where $\tilde{\mathbf{x}}$ is a noise-perturbed version of $\mathbf{x}$, so $q_\sigma(\tilde{\mathbf{x}}|\mathbf{x}) = \mathcal{N}(\mathbf{x}, \sigma)$, and $C$ is a constant depending on $\sigma$ but *not* on $\theta$. When the noise is 0, i.e. $\tilde{\mathbf{x}} = \mathbf{x}$, the left-hand side of the equation above is the explicit score matching loss. Although in theory we can start from extremely small Gaussian noise, and directly optimize the right hand of equation (4), empirically it is beneficial to start with large Gaussian noise and gradually decrease the noise magnitude until $q_\sigma(\mathbf{x}) \approx p(\mathbf{x})$ and $s_\theta(\mathbf{x}) \approx \nabla_{\mathbf{x}}\log q_\sigma(\mathbf{x}) \approx \nabla_{\mathbf{x}}\log p(\mathbf{x})$. Specifically, this has been shown to stabilize the training and improve score estimation [48].

## 3 Methods

### 3.1 Do we really need to match the score?

The Langevin dynamics provides an elegant way to construct an SDE given a specific stationary distribution. However, as noted above, the neural network dynamics seldom have a drift term that is a gradient field (Appendix B). A natural question is therefore whether an SDE with a drift term that is *not* a gradient field (also known as irreversibility) also gives rise to a specific stationary distribution. The answer requires us to look at the Fokker-Planck equation:

$$\partial_t p(\mathbf{v}, t) = \nabla \cdot (\Sigma \nabla p - p F_\theta(\mathbf{v})) \quad (5)$$

where $p(\mathbf{v}, t)$ is the probability density function of the variable of interest $\mathbf{v}$ at time $t$, $F_\theta(\mathbf{v})$ is the drift term of the neural dynamics parametrized by $\theta$, and $\Sigma = \frac{1}{2}\sigma\sigma^T$ is the diffusion matrix. Since the right-hand side of (5) needs to be 0 for a given stationary distribution $p(\mathbf{v})$, we have

$$\nabla \cdot (\Sigma \nabla p - p F_\theta(\mathbf{v})) = 0. \quad (6)$$

Therefore $G := -\Sigma \nabla p + p F_\theta(\mathbf{v})$ needs to be a divergence-free (DF) vector field. In other words, $p F_\theta(\mathbf{v})$ is unique up to a DF field given a fixed stationary distribution $p$. Ma et al. [36] shows that there exists a skew-symmetric matrix $Q$ such that the DF field can be written as $G = Q\nabla p + p\sum_j \frac{\partial}{\partial \mathbf{v}_j}Q_{ij}$, however, this does not shed more light on how expressive $\{F_\theta\}_\theta$ needs to be without more knowledge of $Q$ and its derivative. We show below that under certain conditions, the DF field $G$ can be regarded as a component that is orthogonal to the score function. Therefore the function space $\{F_\theta\}_\theta$ needs to have enough basis functions so that (when projected) it is able to approximate the score function of the target distribution.

We first note that it would be convenient if $F_\theta(\mathbf{v})$ could approximate the gradient fields $\Sigma^{-1}\nabla\log p$ for any $p$, in which case $G = 0$. To find out if this is a necessary condition for the dynamics to sample from an arbitrary target distribution $p$, we let $\Sigma = \mathbf{I}$ and invoke the Helmholtz-Hodge decomposition (HHD) [8, 37]. The decomposition theorem states that any sufficiently smooth vector field in $L^2(\mathbb{R}^n; \mathbb{R}^n)$ can be uniquely decomposed into a DF vector field and a pure gradient field. In other words, the function space of all DF fields $G$ and the function space of all gradient fields $\nabla p$ are the orthogonal complement of each other. Therefore the projection of $p F_\theta(\mathbf{v})$ onto the subspace of smooth gradient fields still needs to be able to approximate $\nabla p$ despite the freedom to choose arbitrary DF field $G$.

For example, in the 1-D case, if we assume that both $p F_\theta(\mathbf{v})$ and $\nabla p$ are square-integrable (so they vanish at infinity), then the divergence-free vector field $G$ (which must be constant in 1-D) is 0, therefore $F_\theta(\mathbf{v}) = \nabla\log p$. As a result, $\{F_\theta\}_\theta$ indeed needs to be able to approximate $\nabla\log p$ for every $p$. In higher dimensions, the same conclusion holds under the assumption of a strict orthogonality constraint:

**Proposition 1.** *Let $p$ be the stationary distribution of the neural dynamics, and the diffusion matrix be the identity matrix. If the DF field $G$ is strictly orthogonal to the gradient field $\nabla p$, meaning that $G(\mathbf{v}) \cdot \nabla p(\mathbf{v}) = 0$ for all $\mathbf{v}$, then the drift term $F_\theta(\mathbf{v})$ can be written as the sum of a divergence-free field $p^{-1}G$ and a gradient field $\nabla\log p$.*

The proof and further detail is in Appendix A. The form of the above decomposition coincides with that of the HHD [8, 9]. Therefore if we enforce the *normal-parallel* boundary condition [15] for the gradient component, the HHD theorem [8] says that the orthogonal projection of the function space $\{F_\theta\}_\theta$ onto the space of gradient fields is the function space of gradient fields $\{\nabla \log p\}_p$ satisfying the boundary condition given the strict orthogonality constraint on $G$. The upshot is that $\{F_\theta\}_\theta$ needs to admit enough basis functions (Appendix A). Note that the boundary condition will not be restrictive if we take a sufficiently large bounded region. Therefore it is essential for the neural dynamics to have an expressive functional form that is able to approximate complex score functions, even if their dynamics are not gradient fields.

Previous work has rigorously established that the strict orthogonality constraint holds, in particular, when the nonlinear dynamics is linearized around fixed points of $\mathbf{v}$ (where the drift term is zero [2, 31, 41]). As a consequence, the conditions of Proposition 1 are true locally around fixed points.

In what follows, we assume that the conditions of the Proposition 1 hold. Under this assumption, without loss of generality, we set $G$ to be $\mathbf{0}$ in the following text and explore whether $\Sigma^{-1}F_\theta$, which is determined by the specific neural dynamics (Figure 1), is able to approximate complex score function $\nabla \log p$.

## 3.2 Synaptic current dynamics: sampler-only networks with limited capacity

We consider the following stochastic synaptic current dynamics [16] (cf. eq. 7.39) that describe a recurrent neural network in terms of the synaptic current that each neuron receives:

$$d\mathbf{v} = D(-\mathbf{v} + W\phi(\mathbf{v}) + I)dt + \sigma d\mathbf{B}_t := F_\theta^{\mathrm{SC}}(\mathbf{v})dt + \sigma d\mathbf{B}_t \tag{7}$$

where $F_\theta^{\mathrm{SC}}(\mathbf{v}) := D(-\mathbf{v} + W\phi(\mathbf{v}) + I)$, $\mathbf{v} = [v_1, \cdots, v_m]^T \in \mathbb{R}^m$ is the synaptic current of the $m$ neurons in the recurrent network, $D \in \mathbb{R}^{m \times m}$ is a diagonal matrix where diagonal elements are the decay constants, $d_i = \tau_i^{-1}$, $W \in \mathbb{R}^{m \times m}$ is the connection matrix, $\phi(\cdot)$ is a nonlinear transfer function[2], $I \in \mathbb{R}^m$ is the external input, $\sigma \in \mathbb{R}^{m \times l}$ is the diffusion coefficient and $\mathbf{B}_t$ is an $l$-dimensional standard Wiener process. The diffusion term can be interpreted as input from other brain areas (due to the large number of incoming connections, the assumption of Gaussianity can be justified by the central limit theorem [21]). We assume that $\theta = \{D, W, I\}$ are tunable parameters through biological learning. To show the limited expressivity of $F_\theta^{\mathrm{SC}}$, we have the following corollary from the Hilbert projection theorem showing that $F_\theta^{\mathrm{SC}}$ is only able to approximate functions in a finite-dimensional function space.

**Proposition 2.** *Let* $H : \mathbb{R}^m \to \mathbb{R}^m$, *a function in the Hilbert space* $L_2(\mathbb{R}^m, \mathbb{R}^m; p)$. *Let* $\Pi$ *be the orthogonal projection operator onto the vector subspace*

$$E = \left\{ A\mathbf{v} + B\phi(\mathbf{v}) + I | A, B \in \mathbb{R}^{m \times m}, I \in \mathbb{R}^{m \times 1} \right\}$$

*If* $\|H - \Pi H\| > 0$, *then* $\inf_\theta \left\| H(\mathbf{v}) - F_\theta^{\mathrm{SC}}(\mathbf{v}) \right\| \geq \|(1 - \Pi)H\| > 0$.

Proof of the proposition is given in Appendix B.1. The proposition says that no matter how the parameter of $F_\theta^{\mathrm{SC}}$ is tuned, the difference between $F_\theta^{\mathrm{SC}}$ and the target function cannot approach 0, and the lower bound of the error is given by the norm of the component in the target function that is orthogonal to the finite-dimensional function space $E$. Therefore, synaptic current dynamics have a limited ability to match the score function and hence limited ability to sample from complex probability distributions under the strict orthogonality constraint in Section 3.1. The conclusion holds even if we let the diffusion coefficient $\sigma$ be tunable. Since $\Sigma^{-1}$ is linear, $\left\{ F_\theta^{\mathrm{SC}} \right\}_\theta$ share the same set of basis functions as $\left\{ \Sigma^{-1}F_\theta^{\mathrm{SC}} \right\}_{\theta,\sigma}$. As we will see in the next section, the firing rate dynamics of a recurrent neural circuit with a separate output layer (a reservoir-sampler network) and a learnable diffusion coefficient $\sigma$ can sample from arbitrary stationary distributions.

## 3.3 Firing-rate dynamics could be a universal sampler

### 3.3.1 Sampler-only networks

In this section, we consider the firing rate dynamics [16] (cf. eq. 7.11) that describe a recurrent neural circuit in terms of firing rates of the neurons. We first consider the sampler-only network:

$$d\mathbf{r} = D(-\mathbf{r} + \phi(W_{\mathrm{rec}}\mathbf{r} + I))dt + \sigma dB_t := F_\theta^{\mathrm{FR}}(\mathbf{v})dt + \sigma d\mathbf{B}_t. \tag{8}$$

---

[2]We later need it to be non-polynomial in Theorem 3

Here, $F_\theta^{\mathrm{FR}}(\mathbf{v}) = D(-\mathbf{r} + \phi(W_{\mathrm{rec}}\mathbf{r} + I))$ and $D$ is a diagonal matrix with decay constants as its diagonal elements. The stationary solution of the corresponding Fokker-Planck equation satisfies

$$\nabla \cdot (\Sigma \nabla p - p F_\theta^{\mathrm{FR}}) = 0 \tag{9}$$

where $\Sigma := \frac{1}{2}\sigma\sigma^T$ is symmetric positive definite (SPD) and $F_\theta^{\mathrm{FR}} = D(-\mathbf{r} + \phi(W_{\mathrm{rec}}\mathbf{r} + I))$. Here $\theta = \{D, W_{\mathrm{rec}}, I\}$ are tunable parameters. If $\Sigma$ is invertible, equation (9) becomes $\nabla \cdot (\Sigma(\nabla p - p\Sigma^{-1}F_\theta^{\mathrm{FR}}))$. Therefore if $\Sigma^{-1}F_\theta^{\mathrm{FR}} = \nabla \log p$ is a gradient field, then the stochastic dynamics of the recurrent neural network described by equation (8) have a stationary distribution $p^*$ such that the score of this distribution $\nabla \log p^* = \Sigma^{-1}F_\theta^{\mathrm{FR}}$.

Compared to the synaptic current dynamics where we could only have functional basis $\mathbf{v}_i$ and $\phi(\mathbf{v})_i$, we can now freely choose the functional basis spanning $\{F_\theta^{\mathrm{FR}}\}_\theta$ depending on $W_{\mathrm{rec}}$ and $I$, but since there is no linear term before the nonlinear transformation $\phi$, the function set $\{F_\theta^{\mathrm{FR}}\}_\theta$ is not closed under addition. If we view $\Sigma^{-1}F_\theta^{\mathrm{FR}}$ as a neural network with one hidden layer, the number of hidden neurons must be the same as the input dimension, and the diffusion matrix $\Sigma$ (hence $\Sigma^{-1}$) is restricted to be an SPD matrix. Therefore, we do not get universal approximation power from $\Sigma^{-1}F_\theta^{\mathrm{FR}}$, and combined with results in Section 3.2, we see that an RNN by itself does not intrinsically produce samples from arbitrary distributions. As we will see below, if we let a population of output neurons receive inputs from a large reservoir of recurrently connected neurons (a reservoir-sampler network), we are able to obtain samples from complex distributions from the output neurons.

### 3.3.2 Reservoir-sampler networks

Now we consider the reservoir-sampler network where there is a linear readout layer $W_{\mathrm{out}} \in \mathbb{R}^{m \times n}$ of the reservoir whose dynamics is given by equation (8) (see also the upper row of Figure 1). As a special case of Ito's lemma, we have

$$\begin{aligned} W_{\mathrm{out}}d\mathbf{r} = dW_{\mathrm{out}}\mathbf{r} &= W_{\mathrm{out}}F_\theta^{\mathrm{FR}}dt + W_{\mathrm{out}}\sigma dB_t \\ &= (-W_{\mathrm{out}}D\mathbf{r} + W_{\mathrm{out}}D\phi(W_{\mathrm{rec}}\mathbf{r} + I))dt + W_{\mathrm{out}}\sigma dB_t. \end{aligned} \tag{10}$$

Now we assume that $W_{\mathrm{rec}}$ is the product of $\widetilde{W}_{\mathrm{rec}}$ and $W_{\mathrm{out}}$, i.e. $W_{\mathrm{rec}} = \widetilde{W}_{\mathrm{rec}}W_{\mathrm{out}}$ and $D = \alpha\mathbf{I}$ is a scaled identity matrix. If we denote the output of the recurrent neural network as $\mathbf{x} := W_{\mathrm{out}}\mathbf{r} \in \mathbb{R}^m$, we derive the following stochastic dynamics for output neurons:

$$d\mathbf{x} = (-\alpha\mathbf{x} + \alpha W_{\mathrm{out}}\phi(\widetilde{W}_{rec}\mathbf{x} + I))dt + W_{\mathrm{out}}\sigma dB_t := \widetilde{F}_\theta^{\mathrm{FR}}(\mathbf{x})dt + \tilde{\sigma}dB_t \tag{11}$$

where $\widetilde{F}_\theta^{\mathrm{FR}}(\mathbf{x}) = (-\alpha\mathbf{x} + \alpha W_{\mathrm{out}}\phi(\widetilde{W}_{rec}\mathbf{x} + I))$ and $\tilde{\sigma} = W_{\mathrm{out}}\sigma$. Therefore in order for the output neurons to sample from a stationary distribution $p$, we need $s_\beta(\mathbf{x}) = (\frac{1}{2}\tilde{\sigma}\tilde{\sigma}^T)^{-1}\widetilde{F}_\theta^{\mathrm{FR}}(\mathbf{x}) := \widetilde{\Sigma}^{-1}\widetilde{F}_\theta^{\mathrm{FR}}(\mathbf{x})$ to match the score $\nabla \log p(\mathbf{x})$. Here $\beta = \{\widetilde{W}_{\mathrm{rec}}, W_{\mathrm{out}}, I, \sigma\}$ are tunable parameters.

Additionally, we assume that $\widetilde{F}_\theta^{\mathrm{FR}}$ is $\mathbf{0}$ outside a reasonable range of $\mathbf{x}$. This assumption is used to prevent $s_\beta(\mathbf{x})$ from behaving wildly outside the bounded region on which $s_\beta(\mathbf{x})$ has the expressivity to match the score. The following theorem proves that with a large enough number of reservoir neurons, the score-matching loss can be arbitrarily small. The proof is given in Appendix C.

**Theorem 3.** *Suppose that we are given a probability distribution with continuously differentiable density function $p(\mathbf{x}) : \mathbb{R}^m \to \mathbb{R}^+$ and score function $\nabla \log p(\mathbf{x})$ for which there exist constants $M_1, M_2, a, k > 0$ such that*

$$p(\mathbf{x}) < M_1 e^{-a\|\mathbf{x}\|} \tag{12}$$

$$\|\nabla \log p(\mathbf{x})\|^2 < M_2 \|\mathbf{x}\|^k \tag{13}$$

*when $\|\mathbf{x}\| > L$ for large enough $L$. Then for any $\varepsilon > 0$, there exists a recurrent neural network whose firing-rate dynamics are given by (11), whose recurrent weights, output weights and the diffusion coefficient are given by $W_{\mathrm{rec}} \in \mathbb{R}^{n \times n}$ of rank $m$, $W_{\mathrm{out}} \in \mathbb{R}^{m \times n}$, and $\sigma \in \mathbb{R}^{n \times m}$ respectively, such that, for a large enough $n$, the score of the stationary distribution of the output units $s_\theta(\mathbf{x})$ satisfies $\mathbb{E}_{\mathbf{x} \sim p(\mathbf{x})}[\|\nabla \log p(\mathbf{x}) - s_\theta(\mathbf{x})\|^2] < \varepsilon$.*

This theorem says that for any realistic data distribution with a smooth positive density function, there always exists a reservoir of recurrently-connected neurons whose output units give samples from a

distribution whose score function approximates that of the data distribution to arbitrary precision. Given the bound on the score matching error, Block et al. [10] (cf. Theorem 13) gives bounds in Wasserstein 2-distance between the stationary distribution of the trained recurrent dynamics and the true data distribution. It is also worth noting that the recurrent weight matrix of the neural circuit in the theorem is of low-rank, so regardless of how many neurons there are in the reservoir, we can always find a low-rank recurrent weight matrix such that the output neurons sample from a correspondingly low-dimensional distribution.

## 3.4   An efficient RNN weight learning algorithm

The training procedure is derived from the proof of Theorem 3. The main idea is to first train an auxiliary neural network with one hidden layer, then transfer the weights of the auxiliary neural network to the weights of the recurrent network that we are considering. More specifically, we first optimize an auxiliary feedforward neural network with one hidden layer $W_{\text{out}}\phi(\widetilde{W}_{\text{rec}}\mathbf{x} + I)$ using backpropagation with the denoising score matching loss (4) such that

$$2\alpha(W_{\text{out}}\phi(\widetilde{W}_{\text{rec}}\mathbf{x} + I) - \mathbf{x}) \approx \nabla \log p(\mathbf{x}). \tag{14}$$

Then we can calculate the diffusion coefficient $\sigma = W_{\text{out}}^T(W_{\text{out}}W_{\text{out}}^T)^{-1}$ and the real recurrent weights $W_{\text{rec}} = \widetilde{W}_{\text{rec}}W_{\text{out}}$ accordingly. The noise magnitude added to the data samples is decreased exponentially over the entire training period. Figure 2 illustrates how the network can gradually learn the score function during training. We refer the readers to Appendix D for more details.

Note that this method of using an auxiliary neural network is much more computationally efficient than directly matching the score of the stationary distribution of the dynamics (11), for which the score function, which involves the matrix inverse $(\tilde{\sigma}\tilde{\sigma}^T)^{-1}$, needs to be recomputed at each optimization step. Furthermore, since the entire training procedure is equivalent to training a feedforward network with one hidden layer, it sidesteps the often challenging temporal computations associated with the Backpropagation Through Time (BPTT) algorithm used to train deterministic RNNs.

Although we assumed that the divergence free field $G = 0$ for the purpose of theoretical analysis, in practice, fast sampling is a major concern for implementing sampling-based inference models of the brain [25, 1, 19, 38], and reversible stochastic dynamics, i.e. if $G = 0$, are known for their slow sampling speed. Fortunately, there is a straightforward way to extend our framework and train RSNs to implement irreversible dynamics with a non-zero divergence-free field $G$. This results in improvements in sampling speed (see Appendix E for details).

## 4   Experimental results

In this section, we present the results from two tasks. First, we try to let the recurrent neural network learn and generate samples from a 1-D double-mode Gaussian mixture distribution and a 2-D mixture of heavy-tailed Laplace distributions. For the second task, we explore whether a reservoir-sample network with firing rate dynamics is able to sample from the distribution of internal representations computed from PCA filtering of image inputs. All dynamics are simulated with the Euler-Maruyama method. See Appendix H for hyperparameters used and other training details.

### 4.1   Learning mixture distributions

We consider a 1-D Gaussian mixture distribution whose density function is the average of two Gaussian distributions centered at $\pm 1$, i.e. $p_{\text{data}}(x) = \frac{1}{2}(\mathcal{N}(-1, 0.25) + \mathcal{N}(1, 0.25))$. We artificially generate 10000 data points from this distribution and minimize the denoising score-matching loss:

$$\mathcal{L}(W_{\text{out}}, \widetilde{W}_{\text{rec}}, I) = \mathbb{E}_{p_{\text{data}}(x)}\mathbb{E}_{\tilde{x}\sim\mathcal{N}(x,\sigma^2)}\left[\left\|2\alpha(W_{\text{out}}\phi(\widetilde{W}_{\text{rec}}x + I) - x) + \frac{\tilde{x} - x}{\sigma^2}\right\|^2\right]. \tag{15}$$

We use $\phi(\cdot) = ReLU(\cdot)$ and set $\alpha = 1/2$. See Figure 2 for the numerical results. It is worth noting that if we use $\tanh$ as the transfer function, the sampler-only networks are able to learn the score function perfectly, as the score function of the Gaussian mixture distribution we considered is exactly spanned by $f_1(x) = x$ and $f_2(x) = \tanh(2x)$. See Appendix F.1 for an example where sampler-only networks fail to learn the score function even if hyperbolic tangent nonlinearity is used.

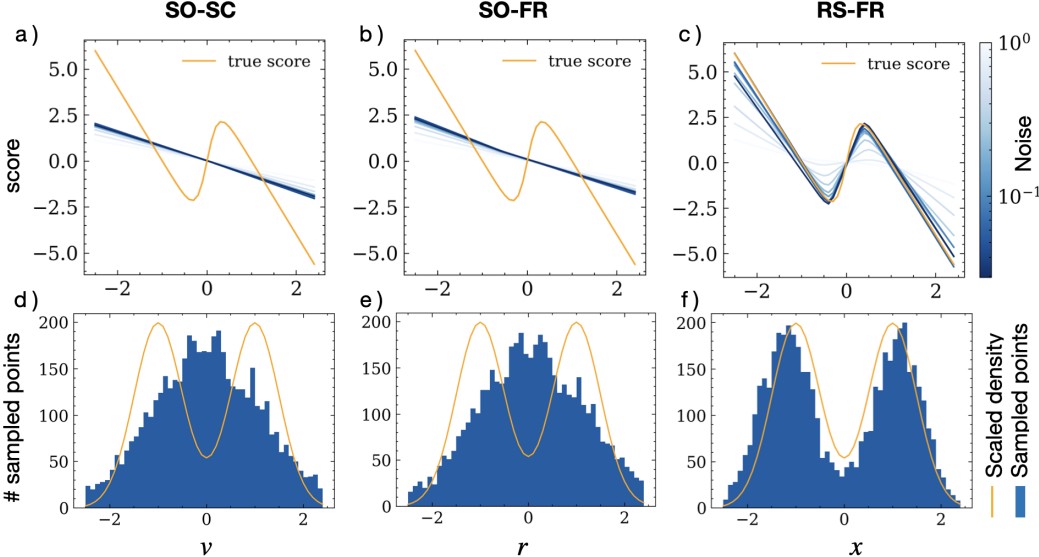

Figure 2: **Bimodal distribution sampling results.** The 3 tractable cases shown are Sampler-only (SO) networks with both synaptic current (SC) dynamics and firing rate (FR) dynamics and Reservoir-sampler (RS) networks with FR dynamics, which are named SO-SC, SO-FR and RS-FR respectively. a-c) The score function learned compared to the true score function (orange curve) as we gradually decrease the noise level (the darker the line, the lower the noise level). We see that RS-FR is capable of perfectly fitting the score function, while SO-SC and SO-FR are only able to fit the score function with piecewise linear functions when using the ReLU transfer function. d-f) Histogram of sampled points, and the (scaled) density function of the target distribution. Again the reservoir-sampler network is able to generate samples whose distribution matches the target distribution, while the sampler-only network is not able to do so due to the incorrectly matched score function.

Next, we show that the model can learn mixtures of heavy-tailed distributions that are evident in natural image statistics and the neural representations in the primary visual cortex [47, 39]. We trained the Reservoir-Sampler network with FR dynamics (RS-FR) model on 20000 sampled data points from a 2-D Laplace mixture distribution, whose density is given by $p_{\text{data}}(\mathbf{x}) = \frac{1}{2} \left( \text{Lap} \left( \mathbf{0}, \begin{bmatrix} 1 & 0.9 \\ 0.9 & 1 \end{bmatrix} \right) + \text{Lap} \left( \mathbf{0}, \begin{bmatrix} 1 & -0.9 \\ -0.9 & 1 \end{bmatrix} \right) \right)$, where Lap denotes the multivariate Laplace distribution. The model successfully learned the probability density of the mixture distribution (Figure 3 left vs. middle), and captured the heavy tails of the distribution as measured by the kurtosis (Figure 3 right).

## 4.2 MNIST generation task

We also tested the sampling ability of our model using the MNIST dataset [32] which contains 60,000 handwritten digits from 0 to 9. We projected MNIST images to a 300-D latent space spanned by the first 300 principal components, and trained the weights of the recurrent neural network as described in Section 3.4 so that the RNN can sample from the latent distribution. To test the model, we generated images by applying inverse PCA projection to samples generated by the model. The schematics and generated images are shown in Figure 4. Note that since we are essentially using a shallow network to match the score, we should not expect comparable performance to generative models that use deep ANNs. Our main goal is to illustrate that the reservoir-sampler network using firing rate dynamics is qualitatively more expressive than other traditional neural sampling models (Appendix G). Finally we also note that it is highly nontrivial for recurrent neural dynamics to complete such a generative task, and to the best of our knowledge, no previous work has achieved such results.

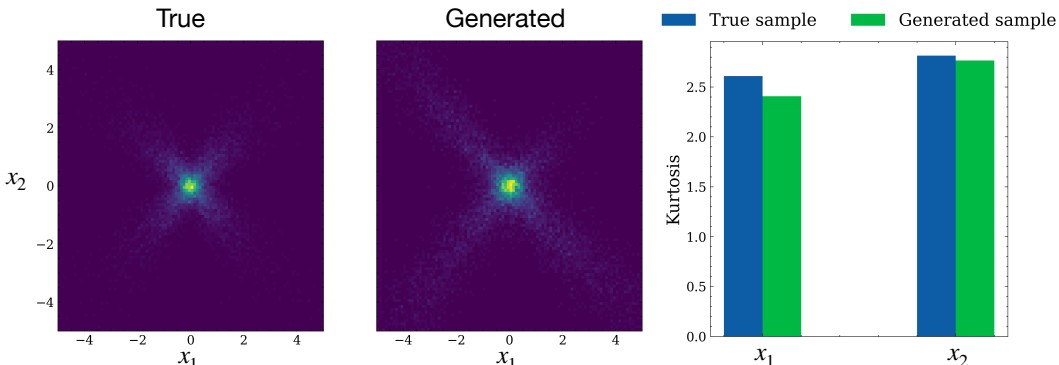

Figure 3: **RS-FR model learning a mixture of 2-D Laplace distribution.** Left to right: sample density from the true distribution (brighter color denotes higher density); sample density from the learned distribution; marginalized kurtosis of each dimension from the true and learned distribution.

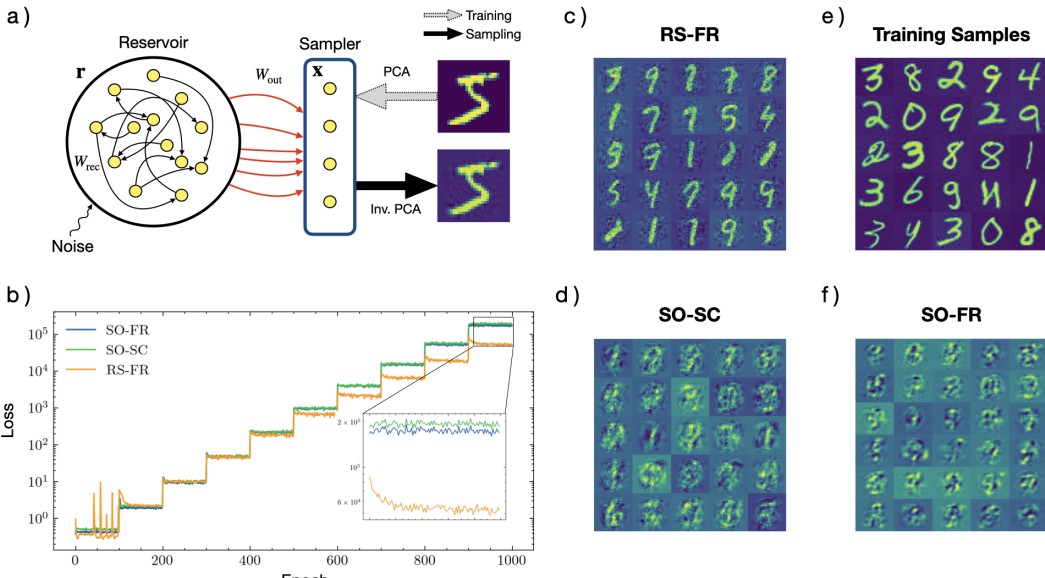

Figure 4: **Learning to sample the MNIST image distribution.** a) An MNIST image is projected to a 300-D latent space (orange circle) spanned by the first 300 principal components using PCA. The sampler learns to sample from the distribution of this latent space and generates images by applying inverse PCA projection to these samples. The diagram illustrates the RS-FR model. b) The loss curves for 3 different models during training. Every 100 epochs the noise level added to the training samples is reduced (Appendix H), and the noise increases to a higher value because the score matching loss magnitude depends on the noise level. As shown in the inset, the loss of the RS-FR model decreases throughout the training process when using the lowest fixed noise level. Meanwhile, the losses of the other two models remain unchanged. c-f) The images generated for the 3 models compared to the digit images generated from latent training samples.

## 5   Discussion

From the perspective of functional analysis and SDE theory, we prove that under the strict orthogonality constraint, it is essential for neural circuits to have a drift term that has the expressivity to approximate complex score functions, despite the fact that the dynamics do not have to exactly implement the Langevin dynamics. We investigated whether a population of neurons can sample from an arbitrary distribution directly and proved that the drift term of the synaptic current dynamics can only sample from a finite-dimensional function space. Although the drift term of the firing

rate dynamics can approximate functions spanned by different basis functions, the number of basis functions is limited. To address this problem, we proposed the reservoir-sampler network for firing rate dynamics. We found that with learnable diffusion coefficients and a sufficiently large reservoir of hidden neurons, the output neurons described using the firing rate dynamics are able to sample from arbitrary data distributions. Our results partly answer the question of what architecture recurrent neural circuits need so that they are able to sample from complex data distributions.

Our analysis and empirical experiments affirm the universality of stochastic RNNs. However, this universality comes with limitations. First, we have only analytically shown the existence of weights that enable sampling from complex data distributions; there is no guarantee that one will find such weights through backpropagation. Additionally, in order to obtain the tunable diffusion coefficient during training, a matrix inverse is needed (likewise in the FORCE algorithm [50]). Further, the question of how biological circuits compute the specific gradient and implement the denoising score-matching algorithm remains an open question. Moreover, in our current formulation, we are only able to approximate the score function with a shallow network with one hidden layer. Our preliminary experiments show that one-hidden-layer RSNs cannot readily approximate high-dimensional heavy-tailed distributions (*e.g.*, those of overcomplete sparse coding representations [39]). It is unclear if this is because of insufficient number of reservoir neurons. Due to the limitation of the GPU memory, we did not try higher number of reservoir neurons.

Our model differs from recent diffusion models [26, 49], which can be seen as time-inhomogeneous SDEs, and has the advantage of being able to run indefinitely in time, making it a suitable candidate for modeling spontaneous activity in the brain. Moreover, while Song and Ermon [48] optimizes the denoising score matching loss at different noise levels jointly, we adopt a sequential learning procedure by gradually decreasing the noise level of the training samples. This procedure is more aligned with the developmental processes involved in forming visual representations in the infant brain, where the distribution of visual representations are thought to be noisier (less linearly separable) initially [4]. Our study therefore serves as a starting point for building a mechanistic model for probabilistic computation in the brain that has similar generative power to current AI generative models.

Biologically, there are multiple ways to interpret the reservoir neurons and sampler neurons in an RSN. First, reservoir and sampler neurons could be seen as different types of neurons in a single brain area, where the dynamics of sampler neurons converge quickly to the equilibrium point. Second, even more straightforwardly, the sampler neurons could be seen as a more separate set of neurons located downstream of the reservoir. We also wish to suggest an alternative interpretation. Biological neural networks are known to have non-synaptic signaling networks (e.g. pervasive neuropeptidergic signaling [3], extensive aminergic signaling [6] or potential extrasynaptic signaling [53]) in addition to the synaptic connectivity that is typically modeled (i.e., via connection weights). We suggest that it is possible that the computations of the reservoir may be implemented by non-synaptic networks, and then "read out" by certain neurons' spikes. This possibility is supported by the recent finding of a low correlation between functional activity and the synaptic ("structural") connectome in C.elegans [53]. Morever, if we only take the structural connectome into consideration, then the resulting model of C. elegans would correspond to the sampler-only network, which, as our theory predicts, will have limited sampling capability.

# 6 Conclusion

In this paper, we explore how a recurrent neural circuit can sample from complex probability distributions, an important functional motif in probabilistic brain models. We start from a basic assumption that the recurrent neural circuit could be described as an SDE. We show that a recurrently-connected neural population by itself has a limited capability to implement stochastic dynamics that can sample from complex data distributions. In contrast, we prove that firing rate dynamics of the output units of a recurrent neural circuit (a reservoir-sampler network) can sample from a richer range of probability distributions. These theoretical results, together with our preliminary experimental results, provide a sufficient condition for neural sampling-based models to exhibit universal sampling capability. Our results therefore provide a foundation for the next generation of sampling-based probabilistic brain models that can explain a wider range of cognitive behaviors.

## 7 Acknowledgements

We are thankful to Profs. Hong Qian, Bamdad Hosseini and Edgar Walker for their guidance and insight with this project. We gratefully acknowledge the support of the grant NIH BRAIN R01 1RF1DA055669.

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

# A Helmholtz-Hodge decomposition and proof of Proposition 1

First we give a proof for Proposition 1:

**Proposition.** *Let $p$ be the stationary distribution of the neural dynamics, and the diffusion matrix be the identity matrix. If the DF field $G$ is strictly orthogonal to the gradient field $\nabla p$, meaning that $G(\mathbf{v}) \cdot \nabla p(\mathbf{v}) = 0$ for all $\mathbf{v}$, then the drift term $F_\theta(\mathbf{v})$ can be written as the sum of a divergence-free field $p^{-1}G$ and a gradient field $\nabla \log p$.*

*Proof.* Since $p$ is the stationary distribution of the neural dynamics, from the Fokker-Planck equation we have

$$F_\theta(\mathbf{v}) = p^{-1}G + \nabla \log p \tag{16}$$

where $G$ is divergence free. Now

$$
\begin{aligned}
\nabla \cdot (p^{-1}G) &= \nabla p^{-1} \cdot G + p^{-1} \nabla \cdot G \\
&= \frac{\nabla p \cdot G}{p^2} && \text{G is divergence free} \\
&= 0. && \text{G and } \nabla p \text{ is pointwise orthogonal}
\end{aligned}
$$

Therefore, $p^{-1}G$ is also divergence free. $\qquad\square$

Now we give a statement of the Helmholtz-Hodge decomposition, which is an application of the Hodge decomposition theorem to vector calculus on $\mathbb{R}^n$ [8, 37].

**Theorem 4** (**Helmoholtz-Hodge Decomposition**). *A smooth vector field $\mathbf{V}$ on a bounded domain of $\mathbb{R}^n$ can be uniquely decomposed into the sum of a gradient field $\nabla u$ that is normal to the boundary, a divergence-free field $\mathbf{v}$ that is parallel to the boundary and a harmonic field $\mathbf{h}$, i.e. $\mathbf{V} = \nabla u + \mathbf{v} + \mathbf{h}$. Moreover, the three components are orthogonal to each other in the $L_2$ sense.*

We note that the form of the decomposition in Proposition 1 coincides with the Helmholtz-Hodge decomposition with a zero harmonic field. Therefore, we can define the orthogonal projection operator $\mathbb{P}$ that projects $F_\theta$ onto its gradient component $\nabla \log p$. Under the strict orthogonality constraint that requires G to be orthogonal to $\nabla p$ *pointwise*, to be able to sample from a broad range of distributions with $\nabla \log p$ orthogonal to the boundary, we need that for all $p$, there exists $G_p$ and $\theta$ such that $F_\theta(\mathbf{v}) = p^{-1}G_p + \nabla \log p$. Applying the projection operator on both sides, we have

$$\mathbb{P}F_\theta(\mathbf{v}) = \mathbb{P}(p^{-1}G_p + \nabla \log p) = \nabla \log p. \tag{17}$$

Moreover, if the function space $\{F_\theta\}_\theta$ admits a set of basis functions $\{e_i\}_i$, i.e. $F_\theta = \sum_i c_i e_i$, then

$$\nabla \log p = \mathbb{P}F_\theta = \sum_i c_i(\mathbb{P}e_i). \tag{18}$$

This means that the space of $\nabla \log p$ needs to be spanned by $\{\mathbb{P}e_i\}_i$, so $\{F_\theta\}_\theta$ should also be spanned by an infinite number of basis functions $\{e_i\}_i$ if we want to sample from a complex class of $\nabla \log p$.

# B Neural dynamics not following gradient fields

For the meaning of notations, see Section 3.2.

**Proposition 5.** *When $m \geq 2$, $F_\theta^{\mathrm{SC}}(\mathbf{v}) := D(-\mathbf{v} + W\phi(\mathbf{v}) + I)$ is a gradient field if and only if $W$ is a zero matrix.*

*Proof.* ( $\implies$ ) If $F_\theta^{\mathrm{SC}}(\mathbf{v}) = \nabla G(\mathbf{v})$ is a gradient field, then

$$\frac{\partial}{\partial v_j}F_\theta^{\mathrm{SC}}(\mathbf{v})_i = \frac{\partial}{\partial v_i}F_\theta^{\mathrm{SC}}(\mathbf{v})_j = \frac{\partial}{\partial v_i \partial v_j}G(\mathbf{v}) \tag{19}$$

because $F_\theta^{\mathrm{SC}}(\mathbf{v})$ is smooth, and we can switch the order of partial differentiation. Written in the form of (7), we have for all $v_i$ and $v_j$ (where $i \neq j$),

$$\frac{\partial}{\partial v_j}\left(-d_i v_i + d_i \sum_{k=1}^{n} W_{ik}\phi(v_k) + d_i I_i\right) = \frac{\partial}{\partial v_i}\left(-d_j v_j + d_j \sum_{k=1}^{n} W_{jk}\phi(v_k) + d_j I_j\right)$$

$$\frac{\partial}{\partial v_j} d_i W_{ij}\phi(v_j) = \frac{\partial}{\partial v_i} d_j W_{ji}\phi(v_i)$$

$$d_i W_{ij}\phi_v(v_j) = d_j W_{ji}\phi_v(v_i),$$

since $v_i$ and $v_j$ are arbitrary and $\phi$ is nonlinear (so the derivative $\phi_v$ is not the same for all $v$), this implies $d_i W_{ij} = d_j W_{ji} = 0$. But the decay constants cannot be 0. Therefore, $W_{ij} = W_{ji} = 0$

( $\Longleftarrow$ ) If $W$ is a zero matrix, then $F_\theta^{\mathrm{SC}}(\mathbf{v}) = \nabla D\left(-\frac{\mathbf{v}\cdot\mathbf{v}}{2} + I \cdot \mathbf{v}\right)$ is a gradient field. $\qquad\square$

### B.1 Hilbert projection theorem and proof of Proposition 2

Here we directly state the Hilbert Projection Theorem. For detailed proof, we refer readers to standard functional analysis textbooks [44].

**Theorem 6.** *(Hilbert Projection Theorem) Given a Hilbert space $H$, and a nonempty closed convex set $E \subset H$, for any $x \in H$, there exists a unique $\Pi x \in E$, such that $\|\Pi x - x\| = \inf_{c \in E} \|x - c\|$.*

Here $\Pi$ is called the orthogonal projection operator. Next, we give the proof of Proposition 2.

*Proof.* Here we consider a special case of a Hilbert space $L_2(\mathbb{R}^m, \mathbb{R}^m; p)$, where the inner product is defined by $\langle f, g \rangle = \int_{\mathbb{R}^m} |f \cdot g| \, p(\mathbf{x}) d\mathbf{x}$ and we let $E = \{A\mathbf{v} + B\phi(\mathbf{v}) + I \mid A, B \in \mathbb{R}^{m \times m}; I \in \mathbb{R}^{n \times 1}\}$. The set of functions $E$ is a closed convex vector subspace of $L_2(\mathbb{R}^m, \mathbb{R}^m; p)$ because $E$ is closed under addition and finite-dimensional. Hence by Hilbert projection theorem, there exists a unique $\Pi H \in E$ such that $\|(1 - \Pi)H\| = \inf_{F \in E} \|H - F\| > 0$. Because $\{F_\theta^{\mathrm{SC}}\}_\theta \subset E$ if the decay matrix $D$ is diagonal, we have

$$\inf_{F_\theta^{\mathrm{SC}}} \left\|H - F_\theta^{\mathrm{SC}}\right\| \geq \inf_{F \in E} \|H - F\| = \|(1 - \Pi)H\| > 0. \tag{20}$$

$\square$

## C Proof of Theorem 3

First, we give a statement of the universal approximation theorem as in [33].

**Theorem C.1.** *(Universal Approximation Theorem) The continuous transfer function $\phi$ is not polynomial if and only if for every $k \in \mathbb{N}, m \in \mathbb{N}$, compact set $U \subset \mathbb{R}^k$, $f \in C(U, \mathbb{R}^m)$, and $\varepsilon > 0$, there exists $n \in \mathbb{N}, W_2 \in \mathbb{R}^{n \times k}, b \in \mathbb{R}^n, W_1 \in \mathbb{R}^{m \times n}$ such that*

$$\sup_{x \in U} \|f(x) - g(x)\| < \varepsilon \tag{21}$$

*where $g(x) = W_1 \phi(W_2 x + b)$. Or equivalently, $\Sigma_n = \mathrm{span}\{\phi(\mathbf{w} \cdot \mathbf{x} + b) : \mathbf{w} \in \mathbb{R}^k, b \in \mathbb{R}\}$ is dense in $C(\mathbb{R}^k)$ if and only if $\phi$ is not polynomial.*

The theorem says that for any transfer function that is not polynomial ($\tanh$ for example), there is a multi-layer neural network with the transfer function that is able to approximate any continuous function to arbitrary precision on a closed and bounded subset of $\mathbb{R}^m$. The proof of Theorem 3 proceeds in three steps:

1. Choose a large enough compact support $U$ for $s_\theta(\mathbf{x})$ such that

$$\int_{\mathbb{R}^m \setminus U} p(\mathbf{x}) \|\nabla \log p(\mathbf{x}) - s_\theta(\mathbf{x})\|^2 \, d\mathbf{x} = \int_{\mathbb{R}^m \setminus U} p(\mathbf{x}) \|\nabla \log p(\mathbf{x})\|^2 \, d\mathbf{x} < \frac{\varepsilon}{2} \tag{22}$$

2. Choose a large enough $n$ and appropriate $\theta$ such that the universal approximation theorem can be applied to show that

$$\int_U p(\mathbf{x}) \|\nabla \log p(\mathbf{x}) - s_\theta(\mathbf{x})\|^2 \, d\mathbf{x} < \frac{\varepsilon}{2} \tag{23}$$

3. Combine equation (22) and (23) so that we have

$$\mathbb{E}_{\mathbf{x}\sim p(\mathbf{x})}[\|\nabla \log p(\mathbf{x}) - s_\theta(\mathbf{x})\|^2] = \int_{\mathbb{R}^m} p(\mathbf{x}) \|\nabla \log p(\mathbf{x}) - s_\theta(\mathbf{x})\|^2 \, d\mathbf{x} < \varepsilon \qquad (24)$$

We begin with two lemmas that will be helpful in the proof of the theorem. Both of them are standard results [3], but we include here for completeness. Let the upper incomplete Gamma function be defined by $\Gamma(s, C) = \int_C^\infty r^{s-1} e^{-r} dr$.

**Lemma C.2.** *If $s$ is a positive integer, then $\Gamma(s, C) = (s-1)! e^{-C} \sum_{i=0}^{s-1} \frac{C^i}{i!}$*

*Proof.* This can be proved by induction. For $s = 1$,

$$\Gamma(1, C) = \int_C^\infty e^{-r} dr = e^{-C} \qquad (25)$$

Therefore the conclusion holds. For $s > 1$, we assume that for $s = k$, the conclusion holds. Then for $s = k + 1$, Through integration by parts, we have

$$\Gamma(k+1, C) = s\Gamma(k, C) + C^k e^{-C} \qquad (26)$$

Therefore

$$\begin{aligned}
\Gamma(k+1, C) &= k(k-1)! e^{-C} \sum_{i=0}^{k-1} \frac{C^i}{i!} + C^k e^{-C} \\
&= k! e^{-C} \left( \frac{C^k}{k!} + \sum_{i=0}^{k-1} \frac{C^i}{i!} \right) \\
&= k! e^{-C} \sum_{i=0}^{k} \frac{C^i}{i!}
\end{aligned} \qquad (27)$$

Therefore the conclusion holds for $s = k + 1$ as well. By induction, it holds for all positive integer $s$. $\qquad \square$

**Lemma C.3.** $\Gamma(s, C) \to 0$ *as* $C \to \infty$.

*Proof.* When $C \to \infty$, $\Gamma(s, C) < \int_C^\infty r^{\lceil s-1 \rceil} e^{-r} dr$. Therefore W.L.O.G., we can assume that $s - 1$ is a positive integer. By lemma C.2,

$$\lim_{C\to\infty} \Gamma(s, C) = (s-1)! \sum_{i=0}^{s-1} \lim_{C\to\infty} \frac{e^{-C} C^i}{i!} = 0 \qquad (28)$$

since the exponential term $e^{-C}$ is beyond all orders. $\qquad \square$

Now we give a formal proof of Theorem 3.

**Theorem.** *Suppose that we are given a probability distribution with continuously differentiable density function $p(\mathbf{x}) : \mathbb{R}^m \to \mathbb{R}^+$ and score function $\nabla \log p(\mathbf{x})$ for which there exist constants $M_1, M_2, a, k > 0$ such that*

$$p(\mathbf{x}) < M_1 e^{-a\|\mathbf{x}\|} \qquad (29)$$

$$\|\nabla \log p(\mathbf{x})\|^2 < M_2 \|\mathbf{x}\|^k \qquad (30)$$

*when $\|\mathbf{x}\| > L$ for large enough $L$. Then for any $\varepsilon > 0$, there exists a recurrent neural network whose firing-rate dynamics are given by (11), whose recurrent weights, output weights and the diffusion coefficient are given by $W_{\mathrm{rec}} \in \mathbb{R}^{n \times n}$ of rank $m$, $W_{\mathrm{out}} \in \mathbb{R}^{m \times n}$, and $\sigma \in \mathbb{R}^{n \times m}$ respectively, such that, for a large enough $n$, the score of the stationary distribution of the output units $s_\theta(\mathbf{x})$ satisfies $\mathbb{E}_{\mathbf{x}\sim p(\mathbf{x})}[\|\nabla \log p(\mathbf{x}) - s_\theta(\mathbf{x})\|^2] < \varepsilon$.*

---

[3] see https://dlmf.nist.gov/8.11

*Proof.*

**Step 1** First of all, we would like to find a compact set $U_C = \{\mathbf{x} | \|\mathbf{x}\|_2 \leq C\}$ such that $\int_{\mathbb{R}^m \setminus U_C} \|\nabla \log p(\mathbf{x}) - s_\theta(\mathbf{x})\|^2 p(\mathbf{x}) d\mathbf{x} < \frac{\varepsilon}{2}$. We can upper bound the integral by the following calculation using spherical coordinates when $C > L$.

$$\int_{\mathbb{R}^m \setminus U_C} \|\nabla \log p(\mathbf{x}) - s_\theta(\mathbf{x})\|^2 p(\mathbf{x}) d\mathbf{x} = \int_{\mathbb{R}^m \setminus U_C} \|\nabla \log p(\mathbf{x})\|^2 p(\mathbf{x}) d\mathbf{x} \tag{31}$$

$$< \int_{\mathbb{R}^m \setminus U_C} M_1 M_2 e^{-a\|\mathbf{x}\|} \|\mathbf{x}\|^k d\mathbf{x} \tag{32}$$

$$= \frac{2M_1 M_2 \pi^{m/2}}{\Gamma(\frac{m}{2})} \int_C^\infty r^{m+k-1} e^{-ar} dr \tag{33}$$

$$= \frac{2M_1 M_2 \pi^{m/2}}{a^{m+k}\Gamma(\frac{m}{2})} \int_{aC}^\infty u^{m+k-1} e^{-u} du \tag{34}$$

$$= \frac{2M_1 M_2 \pi^{m/2}}{a^{m+k}\Gamma(\frac{m}{2})} \Gamma(m+k, aC) \tag{35}$$

Note that we assume $s_\theta(\mathbf{x}) = \mathbf{0}$ outside of $U_C$ [4]. By lemma C.3, there exists a large enough constant $\widetilde{C}$ such that $\Gamma(m+k, a\widetilde{C}) < \frac{a^{m+k}\Gamma(\frac{m}{2})\varepsilon}{4M_1 M_2 \pi^{m/2}}$. Therefore

$$\int_{\mathbb{R}^m \setminus U_{\widetilde{C}}} \|\nabla \log p(\mathbf{x}) - s_\theta(\mathbf{x})\|^2 p(\mathbf{x}) d\mathbf{x} < \frac{2M_1 M_2 \pi^{m/2}}{a^{m+k}\Gamma(\frac{m}{2})} \cdot \frac{a^{m+k}\Gamma(\frac{m}{2})\varepsilon}{4M_1 M_2 \pi^{m/2}} = \frac{\varepsilon}{2} \tag{36}$$

**Step 2** Next we write down the specific form of $s_\theta(\mathbf{x})$, the score of the stationary distribution of $\mathbf{x}$ for the reservoir-sampler dynamics (11)

$$s_\theta(\mathbf{x}) = 2(W_{\text{out}} \sigma \sigma^T W_{\text{out}}^T)^{-1}(-\alpha \mathbf{x} + \alpha W_{\text{out}} \phi(\widetilde{W}_{\text{rec}} \mathbf{x} + I)). \tag{37}$$

Since $\sigma$ is learnable, we can define $\sigma = W_{\text{out}}^T(W_{\text{out}} W_{\text{out}}^T)^{-1}$ such that $W_{\text{out}} \sigma \sigma^T W_{\text{out}}^T$ is an identity matrix [5]:

$$W_{\text{out}} \sigma \sigma^T W_{\text{out}}^T = W_{\text{out}} W_{\text{out}}^T (W_{\text{out}} W_{\text{out}}^T)^{-1} (W_{\text{out}} W_{\text{out}}^T)^{-1} W_{\text{out}} W_{\text{out}}^T$$
$$= \mathbf{I} \tag{38}$$

As $p$ is continuously differentiable, by Theorem C.1, there exists $W_{\text{out}}, \widetilde{W}_{\text{rec}}$ such that

$$\sup_{\mathbf{x} \in U_{\widetilde{C}}} \left\| W_{\text{out}} \phi(\widetilde{W}_{\text{rec}} \mathbf{x} + I) - \left( \mathbf{x} + \frac{\nabla \log p(\mathbf{x})}{2\alpha} \right) \right\|^2 < \frac{\varepsilon}{8\alpha^2} \tag{39}$$

Therefore with those choices of $W_{\text{out}}, \widetilde{W}_{\text{rec}}$ and $\sigma$, we have

$$\int_{U_{\widetilde{C}}} \|\nabla \log p(\mathbf{x}) - s_\theta(\mathbf{x})\|^2 p(\mathbf{x}) d\mathbf{x} < \sup_{\mathbf{x} \in U_{\widetilde{C}}} \|\nabla \log p(\mathbf{x}) - s_\theta(\mathbf{x})\|^2$$

$$= \sup_{\mathbf{x} \in U_{\widetilde{C}}} \left\| \nabla \log p(\mathbf{x}) - 2(-\alpha \mathbf{x} + \alpha W_{\text{out}} \phi(\widetilde{W}_{\text{rec}} \mathbf{x} + I)) \right\|^2$$

$$= 4\alpha^2 \sup_{\mathbf{x} \in U_{\widetilde{C}}} \left\| \frac{\nabla \log p(\mathbf{x})}{2\alpha} + \mathbf{x} - W_{\text{out}} \phi(\widetilde{W}_{\text{rec}} \mathbf{x} + I) \right\|^2$$

$$< 4\alpha^2 \cdot \frac{\varepsilon}{8\alpha^2}$$

$$= \frac{\varepsilon}{2} \tag{40}$$

---

[4] Equality (31) is the only place where this assumption is used. The equality also holds if $s_\theta(\mathbf{x})$ grows at a similar rate to $\nabla \log p$.

[5] A technical subtlety is that $W_{\text{out}} W_{\text{out}}^T$ is not necessarily invertible. However, since every transformation considered here is continuous, we could add a small perturbation to $W_{\text{out}}$ such that $W_{\text{out}} W_{\text{out}}^T$ is invertible, and our result still holds.

**Step 3**

$$\mathbb{E}_{\mathbf{x}\sim p(\mathbf{x})}[\|\nabla \log p(\mathbf{x}) - s_\theta(\mathbf{x})\|^2] = \left(\int_{\mathbb{R}^m \setminus U_{\widetilde{C}}} + \int_{U_{\widetilde{C}}}\right) p(\mathbf{x}) \|\nabla \log p(\mathbf{x}) - s_\theta(\mathbf{x})\|^2 \, d\mathbf{x}$$
$$< \frac{\varepsilon}{2} + \frac{\varepsilon}{2} = \varepsilon$$

$\square$

## D  Pseudocode for training the reservoir-sampler network

---
**Algorithm 1:** Training RSN

---
**Input:** Training samples $\{\mathbf{x}_i\}$, pseudo recurrent weights $\widetilde{W}_{\text{rec}}$, output weights $W_{\text{out}}$, start
noise level $\sigma_1$, noise decay factor $C$, number of noise level $N$

**Output:** trained recurrent weights $W_{\text{rec}}$, trained output weights $W_{\text{out}}$, trained diffussion
coefficient $\sigma$

**for** $i = 1, \ldots, N$ **do**

    **while** *not done* **do**

        Sample $\mathbf{x}$ from the training set

        Perturb the samples with Gaussian noise $\widetilde{\mathbf{x}} = \mathbf{x}_j + \varepsilon, \varepsilon \sim \mathcal{N}(0, \sigma_i^2 \mathcal{I})$

        Compute the gradient of the score-matching loss

        $\frac{1}{2}\nabla_{\widetilde{W}_{\text{rec}},W_{\text{out}}} \left\|2\alpha(W_{\text{out}}\phi(\widetilde{W}_{\text{rec}}\mathbf{x} + I) - \mathbf{x}) - \frac{\mathbf{x}-\widetilde{\mathbf{x}}}{\sigma_i^2}\right\|^2$

        Update $\widetilde{W}_{\text{rec}}, W_{\text{out}}$ with gradient descent

    $\sigma_{i+1} = C\sigma_i$

$W_{\text{rec}} \leftarrow \widetilde{W}_{\text{rec}}W_{\text{out}}$

$\sigma \leftarrow W_{\text{out}}^T (W_{\text{out}}W_{\text{out}}^T)^{-1}$

**return** $\widetilde{W}_{\text{rec}}, W_{\text{out}}$

---

## E  Accelerated sampling with reservoir-sampler network

To construct an irreversible dynamics that accelerates sampling, we let the divergence-free (DF) field
G be of the form $J\nabla p$, where $J = -J^T$ can be any skew-symmetric matrix that is either learned
through training or prescribed. Then from equation (6) we see that the drift term $F_\theta$ satisfies

$$F_\theta(\mathbf{v}) = \Sigma \nabla \log p(\mathbf{v}) + J\nabla \log p(\mathbf{v}) = (\Sigma + J)\nabla \log p(\mathbf{v}). \tag{41}$$

Therefore the corresponding score matching problem is $\alpha(\Sigma + J)^{-1}(W_{out}\phi(\tilde{W}_{rec}x + I) - x) \approx \nabla \log p(x)$. With a slight modification, the pseudo code for an accelerated RSN training algorithm is
given in algorithm 2. We repeat the experiment in Section 4.1 with the accelerated RSN. The results
are shown in Figure 5. We see that the accelerated RSN is able to sample from the distribution faster
than the RSN.

## F  Additional experiments

### F.1  Failure cases for sampler-only networks with hyperbolic tangent nonlinearity

Figure 6 shows that with tanh nonlinearity, sampler-only network is able to sample from the distri-
bution we consider in Section 4.1. However, this is because the score function of the distribution
considered can be spanned by just two basis functions equipped by the drift term of the sampler-only
network. Therefore, our theory predicts that the sampler-only network will not be able to approximate
score functions that are not exactly spanned by these two basis functions. Indeed, if we consider a
slightly different Gaussian mixture distribution, $p_{\text{data}}(x) = \frac{1}{2}(\mathcal{N}(-1, 0.0625) + \mathcal{N}(1, 0.25))$, then
the sampler-only network will not be able to recover the score function as shown in Figure 7, while
RSN still succeeds in recovering the score function.

---

**Algorithm 2:** Training accelerated RSN

---

**Input:** Training samples $\{\mathbf{x}_i\}$, pseudo recurrent weights $\widetilde{W}_{\text{rec}}$, output weights $W_{\text{out}}$, pseudo
diffusion coefficient $\tilde{\sigma}$, skew-symmetric matrix $J$, start noise level $\sigma_1$, noise decay
factor $C$, number of noise level $N$

**Output:** trained recurrent weights $W_{\text{rec}}$, trained output weights $W_{\text{out}}$, trained diffussion
coefficient $\sigma$

**for** $i = 1, \ldots, N$ **do**

    **while** *not done* **do**

        Sample $\mathbf{x}$ from the training set

        Perturb the samples with Gaussian noise $\widetilde{\mathbf{x}} = \mathbf{x}_j + \varepsilon, \varepsilon \sim \mathcal{N}(0, \sigma_i^2 \mathcal{I})$

        Compute the gradient of the score-matching loss

$$\frac{1}{2} \nabla_{\widetilde{W}_{\text{rec}}, W_{\text{out}}, \tilde{\sigma}, J} \left\| \alpha(\tfrac{1}{2}\tilde{\sigma}\tilde{\sigma}^T + J)^{-1}(W_{\text{out}}\phi(\widetilde{W}_{\text{rec}}\mathbf{x} + I) - \mathbf{x}) - \frac{\mathbf{x}-\widetilde{\mathbf{x}}}{\sigma_i^2} \right\|^2$$

        Update $\widetilde{W}_{\text{rec}}, W_{\text{out}}, \tilde{\sigma}$ with gradient descent

        Update $J$ with gradient descent (Optional)

    $\sigma_{i+1} = C\sigma_i$

$W_{\text{rec}} \leftarrow \widetilde{W}_{\text{rec}} W_{\text{out}}$

$\sigma \leftarrow W_{\text{out}}^T (W_{\text{out}} W_{\text{out}}^T)^{-1} \tilde{\sigma}$

**return** $\widetilde{W}_{\text{rec}}, W_{\text{out}}$

---

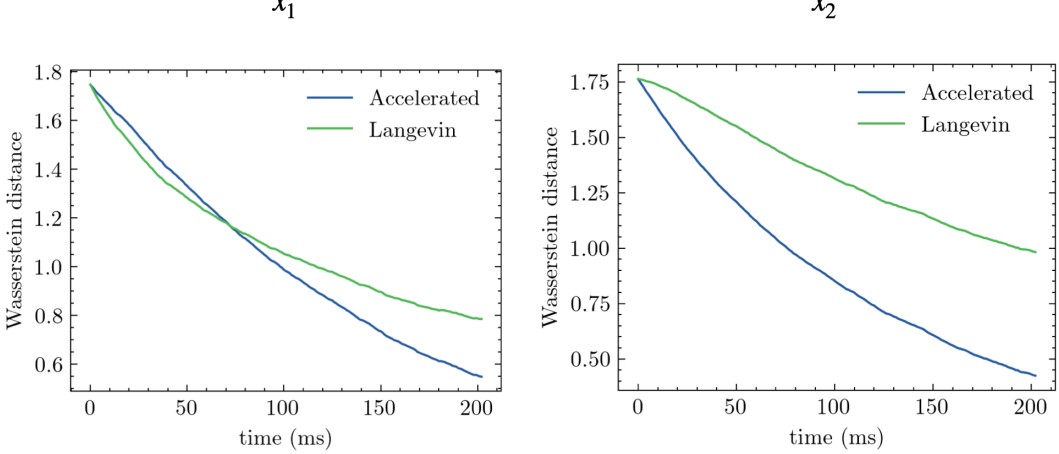

Figure 5: **Sampling speed of accelerated RSN** We repeat the experiment described in Section 4.1 where we try to sample from a 2-D Laplacian mixture distribution. The figure shows the Wasserstein distance between the marginalized distributions of generated samples and true samples for both dimensions. We see that compared to vanilla RSN, accelerated RSN is able to sample from the stationary distribution with faster relaxation time.

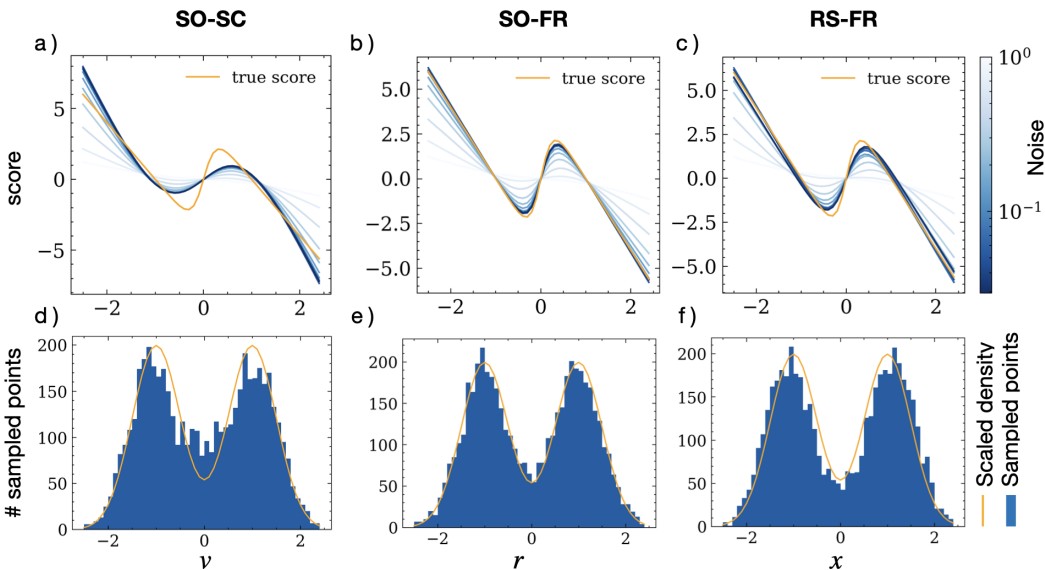

Figure 6: **Sampler-only networks with hyperbolic tangent nonlinearity are able to sample from a 1-D Gaussian mixture distribution with a symmetric score function.**. See Figure 2 for an explanation of the abbreviations. Here we use tanh nonlinearity for sampler-only networks (SO-SC and SO-FR). All 3 types of networks are able to match the score function and sample from the distribution.

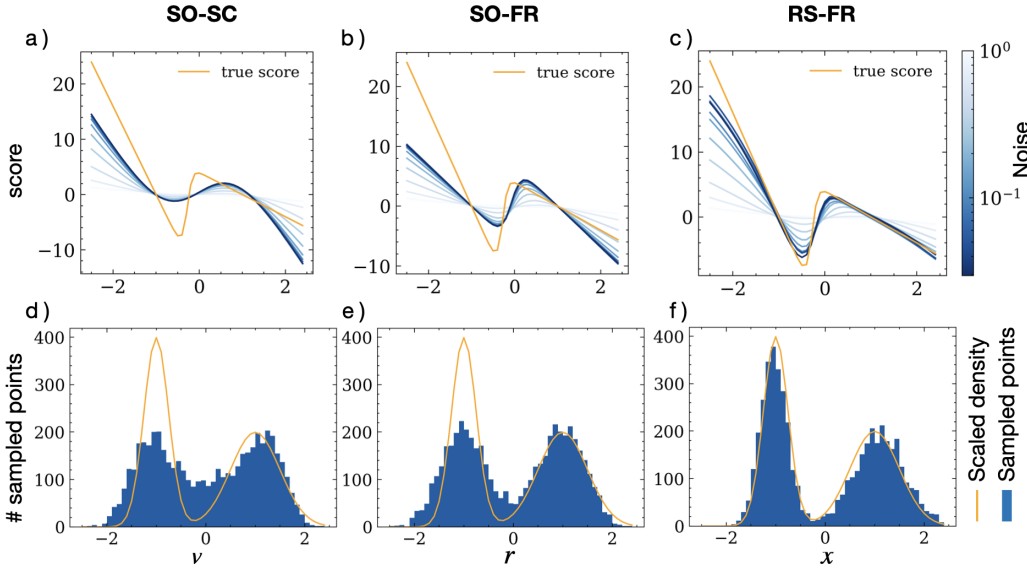

Figure 7: **Sampler-only network with hyperbolic tangent nonlinearity fails to sample from a 1-D Gaussian mixture distribution with a non-symmetric score function.** See Figure 2 for an explanation of the abbreviations. Here we use tanh nonlinearity for sampler-only networks (SO-SC and SO-FR). Only RSN is able to match the score function and sample from the distribution.

### F.2 Sampling using autoencoder

We show that we can replace the fixed linear PCA projection used in Section 4.2 with pretrained nonlinear autoencoder. We test the algorithm on both the MNIST dataset (Figure 8) and the CIFAR-10 dataset (Figure 9).

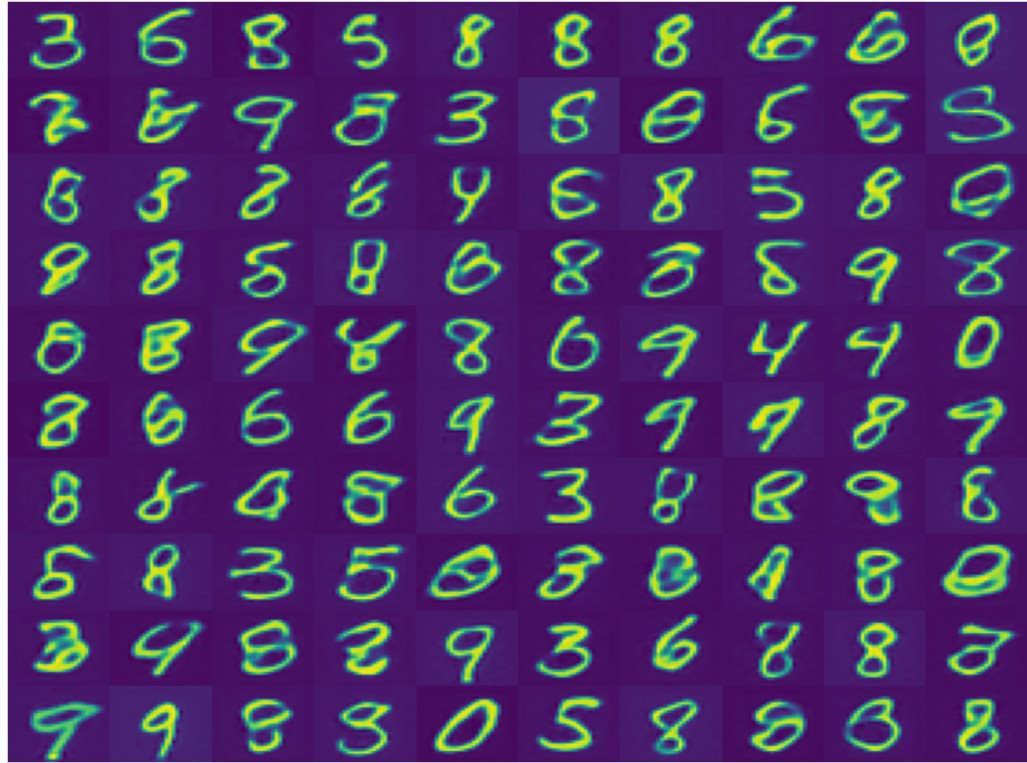

Figure 8: Samples produced by our reservoir sampler network when trained to fit the latent distribution produced by a non-linear convolution autoencoder from MNIST. The distribution was 32-dimensional. The reservoir sampler network had 500 reservoir neurons.

## G   Related works

In this section, we examine the neural dynamics considered in [17, 1, 38, 19] and discuss how our work is related.

Dong et al. [17] uses Hamiltonian Langevin dynamics to accelerate sampling in continuous attractor networks. The Hamiltonian dynamics below are considered in the paper (cf. eq (19) and (20)):

$$\begin{cases} \tau_s \frac{d\mathbf{s}}{dt} & = \boldsymbol{\alpha}\mathbf{y} \\ \tau_z \frac{d\mathbf{y}}{dt} & = -\boldsymbol{\beta}\boldsymbol{\alpha}^{-1}\mathbf{y} + \Lambda(\mathbf{s^o} - \mathbf{s}) + \sqrt{\tau_z}\boldsymbol{\sigma_y}\boldsymbol{\xi}_t. \end{cases} \tag{42}$$

The overdamped Langevin dynamics that samples from the same marginal stationary distribution over $\mathbf{s}$ as (42):

$$\frac{d\mathbf{s}}{dt} = \Lambda(\mathbf{s^o} - \mathbf{s}) + \sqrt{2}\boldsymbol{\xi}_t, \tag{43}$$

where $\mathbf{s^o}$ is an observation generated by a latent feature $\mathbf{s}$ according to a Gaussian distribution, and the linear term $\Lambda(\mathbf{s^o} - \mathbf{s})$ comes from the external input. So the network dynamics in Dong et al. [17] themselves do not generate samples without the external input, and the study does not clarify how to supply the network with nonlinear score functions.

CIFAR 10 Images

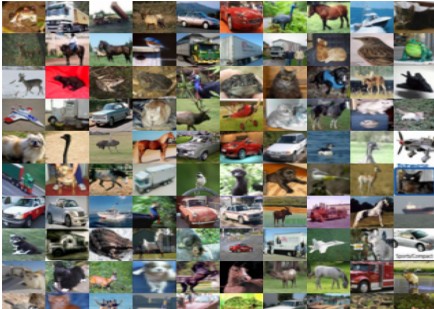

Reconstructed Images

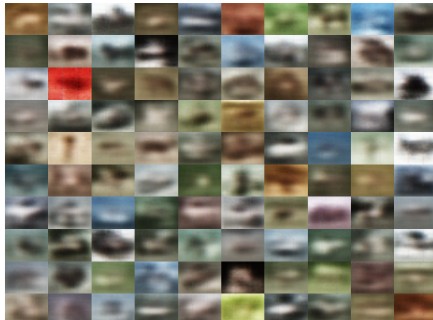

Samples

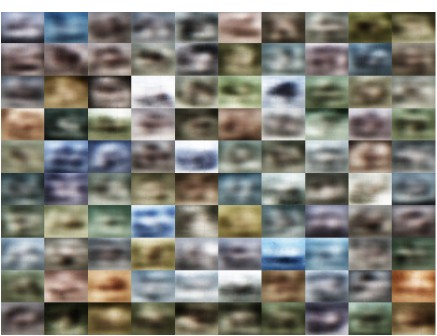

Figure 9: Samples produced by our reservoir sampler network when trained to fit the latent distribution produced by a non-linear convolution autoencoder from CIFAR-10. We show both the original images and the reconstructed images to evaluate the sampling quality. The distribution was 128-dimensional. The reservoir sampler network had 20000 reservoir neurons.

Aitchison and Lengyel [1] have shown that Hamiltonian Monte Carlo (HMC) naturally maps onto the dynamics of excitatory-inhibitory neural networks. However, the neural dynamics considered are entirely linear in terms of the network variables (cf. eq (23) and (24)). Therefore, the dynamics are again equivalent to the synaptic current dynamics with an identity transfer function in Section 3.2. The same applies to the analysis in Masset et al. [38], which also only considers linear neural dynamics. By Proposition 2 and the decomposition discussed in Kwon et al. [31], these dynamics can only sample from probability distributions whose score functions are linear, i.e. Gaussians.

The work by Echeveste et al. [19] provides a unifying model for several dynamical phenomena in sensory cortices by training a nonlinear recurrent excitatory-inhibitory neural circuit model to perform sampling-based probabilistic inference. The synaptic neural dynamics below (cf. equation (8)) is used, where the nonlinear transfer function is supralinear, i.e. $\phi(u) = k\lceil u \rceil^n$:

$$\tau_i \frac{du_i}{dt} = -u_i + \sum_{j=1}^{N} w_{ij}\phi(u_j) + h_i(t) + \eta_i(t). \tag{44}$$

As discussed in Section 3.2, while they are richer than linear dynamics, dynamics of this form still have limited ability to sample from complex probability distributions. Moreover, we note that the training criterion considered in Echeveste et al. [19] only matches the mean and the variance of the stationary distribution. These points are in contrast to our work here, where we show that the reservoir-sampler network, which extends the dynamics (44), can be trained via alternate methods to sample from complex probability distributions by matching their score functions.

# H  Details of numerical experiments

All experiments were run on one NVIDIA Quadro RTX 6000 GPU. We always use the model saved at the last training iteration.

## H.1  Training details for Section 4.1

For the 1-D bimodal Gaussian distribution experiment (Figure 2), we used 1000 reservoir neurons (only in the reservoir sampler network) and 1 sampler neuron. We trained all 3 networks (SO-FR, SO-SC, RS-FR) for 79000 iterations with the Adam optimizer. The learning rate was 0.0001, and the batch size was 128. We started by adding Gaussian noise with standard deviation (std) $\sigma_1 = 1$ to the training samples. We decreased the std every 7900 iterations by a constant factor, i.e. $\sigma_{i+1}/\sigma_i = C$, such that $\sigma_{10} = .01$. When simulating the stochastic neural dynamics, we used a step size of $10^{-4}$, and ran 10000 steps.

For the mixture of 2-D Laplace distributions experiment (Figure 3), we used 1000 reservoir neurons (only in the reservoir sampler network) and 2 sampler neurons. We trained the RS-FR network for 62,500 iterations with the Adam optimizer. The learning rate was 0.0001, and the batch size was 128. We started by adding Gaussian noise with std $\sigma_1 = 1$ to the training samples. We decreased the std every 6250 iterations by a constant factor, i.e. $\sigma_{i+1}/\sigma_i = C$, such that $\sigma_{10} = .01$. When simulating the stochastic neural dynamics, we used a step size of $10^{-4}$, and ran 20000 steps.

## H.2  Training details for Section 4.2

For the MNIST experiment (Figure 2), we used 20000 reservoir neurons (only in the reservoir sampler network) and 300 sampler neurons. We trained all 3 networks (SO-FR, SO-SC, RS-FR) for 1000 epochs with the Adam optimizer. The learning rate was 0.001, and the batch size was 64. We started by adding Gaussian noise with std $\sigma_1 = 1$ to the training samples. We decreased the std every 100 epochs by a constant factor, i.e. $\sigma_{i+1}/\sigma_i = C$, such that $\sigma_{10} = .01$. When simulating the stochastic neural dynamics, we used a step size of $10^{-6}$, and ran 30000 steps.

# I  Statement of code availability

All code is available on Github

