# OpenReview forum: "Expressive probabilistic sampling in recurrent neural networks"
_NeurIPS.cc/2023/Conference — NeurIPS 2023 poster_

### Official Review · Reviewer_pX3M · 2023-06-16

**Soundness:** 2 fair
**Presentation:** 3 good
**Contribution:** 2 fair
**Rating:** 5
**Confidence:** 5

**Summary:**

This paper studies circuit algorithms for sampling-based Bayesian inference in continuous-time rate-based recurrent neural networks. Specifically, it argues that using a linear readout from a noisy reservoir provides a substantial expressivity benefit relative to representing the sampled distribution directly in the recurrent population.

**Strengths:**

1. The question of the minimal recurrent circuit architectures that allow sampling from complex distributions is clearly relevant to the neuroscience community at NeurIPS, and I think the main result is interesting (albeit unsurprising).

2. The experiments show to a reasonable degree that the proposed method can sample from non-trivial non-Gaussian distributions (MNIST). This is an improvement on most comp neuro papers on sampling, which largely focus on Gaussian distributions. My enthusiasm for this point is, however, dampened a bit by the fact that the networks here are rate-based and the authors aren't trying to optimize functionals of the dynamics (e.g. as in work from the Lengyel group), so it's not very surprising that this can be made to work.

**Weaknesses:**

1. How does Proposition 1 differ from the results of Ma, Chen, and Fox, "A Complete Recipe for Stochastic Gradient MCMC" (NeurIPS 2015)? Perhaps I've missed something, but I do not see a novel contribution in Section 3.1.

2. The present manuscript does not address the speed of relaxation to the stationary distribution. The ability to obtain accurate samples quickly is a necessary feature of any model of sampling-based inference in the brain, and indeed the question of how to achieve fast sampling in biologically-plausible implementations has been a major focus of past work (e.g., Hennequin et al. 2014, Aitchison and Lengyel 2016, Echeveste et al. 2020, Masset et al. 2022). In my view, this is a key shortcoming of the learning algorithm proposed in Section 3.4: no attention is paid to whether the resulting network enjoys favorable convergence properties. Of course, one could modify the resulting drift and diffusion terms (as in Hennequin et al. 2014 or Masset et al. 2022), but this would require an additional optimization step. The authors should at least discuss this point; adding more experiments to probe this issue would make for a stronger paper.

3. In Lines 293-295 of the Discussion, the authors state that "Our preliminary experiments show that one-hidden-layer RSNs cannot readily approximate high-dimensional heavy-tailed distributions (e.g., those of overcomplete sparse coding representations [36])." If the authors have such results, they should show them. In particular, related to point (2) above, they should show how the convergence speed and number of reservoir neurons required to obtain accurate samples scales with the dimension of the target distribution. As the authors themselves discuss, their approach is currently limited to approximating the score with a single-hidden-layer MLP, and such Cybenko-Hornik-type approximation results may require immensely wide networks. This is also a limitation of the MNIST experiment: MNIST is intrinsically not that high-dimensional, and already here the authors used a network with 20000 reservoir neurons (Line 638)!

4. I think it's important to mention that the striking result of Figure 2 depends critically on the choice of ReLU activation function, particularly as much of the paper is concerned with existence results. If instead one used $\tanh$, then the performance gap between the trained sampler-only firing rate and reservoir firing rate models should be small. In particular, a sampler-only firing rate network can implement exact Langevin sampling in this case. The score of $p(x) = \frac{1}{2} \mathcal{N}(\mu , \sigma^{2}) + \frac{1}{2} \mathcal{N}(-\mu,\sigma^2)$ is $\frac{d}{dx} \log p(x) = \frac{1}{\sigma^{2}} \left[ - x + \mu \tanh\left(\frac{\mu}{\sigma^{2}} x\right) \right]$. This gives the Langevin dynamics $dx(t) = \frac{d \log p}{dx} dt + \sqrt{2} dB(t) = \frac{1}{\sigma^{2}} \left[ - x + \mu \tanh\left(\frac{\mu}{\sigma^{2}} x\right) \right] dt + \sqrt{2} dB(t)$. With $\mu = 1$ as is used in Figure 2, this is exactly of the form of a single-neuron sampler-only firing rate network in eq (8) of the submitted manuscript, with $D = 1/\sigma^{2}$, $W_{rec} = 1/\sigma^{2}$, $I=0$, and $\phi(x) = \tanh(x)$. One can easily verify numerically that this network produces accurate samples from the desired bimodal distribution.

**Questions:**

- The figure axis labels are too small, making them hard to read. Figure 4b is also too small. Please increase the font and panel sizes to improve legibility.

- Lines 32-33: Along with the other references on Gaussian sampling, Hennequin, Aitchison, and Lengyel, "Fast Sampling-Based Inference in Balanced Neuronal Networks" (NeurIPS 2014), should be cited.

- Line 225: I do not see how the proof of Theorem 3 is "constructive," since it is based on an existence-type universal approximation theorem.

- Lines 303-304: "This procedure is more aligned with the developmental processes involved in forming visual representations in the infant brain, where visual representations are thought to be noisier (less linearly separable) initially [3]" Please elaborate; I don't quite follow.

- Line 317-318: This is quite vague. Having a clearer proposal for what this model might help explain in neuroscience would make for a much stronger paper.

- It would be helpful to repeat the theorem statements in the Supplement.

- Lemmas C.2 and C.3 can be replaced by using standard results on the asymptotic expansion of the upper incomplete Gamma function for large argument, see https://dlmf.nist.gov/8.11.

- In Line 562, in the proof of Theorem 3, should $m$ be replaced by $n$?

**Limitations:**

The authors provide a generally adequate discussion of the limitations of their work (modulo the weaknesses discussed above), but fail to show the data for some points (see point 3 under Weaknesses).

---

> ### Author Rebuttal · Authors · 2023-08-09
>
> We appreciate the reviewer’s critiques and suggestions. Below we address the reviewer’s concerns point by point:
>
> Strengths:
>
> With respect to the two enthusiasm-limiting aspects, we believe we understand, and in our revision will add text specifically pointing these limitations out, with citations to the relevant literature (e.g. Echeveste et al. 2020 and allied papers) as well as the explicit suggestions that future work should extend our findings to these settings.  In more detail, we will explain: First, it is true that we study the ability to produce samples from given “target” distributions, as opposed to studying (input-dependent) networks that additionally solve an inference problem (as in Echeveste et al. 2020).  Second, we study this problem in the context of continuous-valued “rate” networks as opposed to jump-discontinuous spiking systems.
>
> This said, we do believe that our results, which are novel in their scope of general distributions and fairly complex technically, are best presented in terms of sampling from fixed, general (beyond-gaussian)  target distributions and in terms of the broadly used, and mathematically tractable rate networks.  (With respect to a tractable partial extension towards spiking networks, please also see our response to Q3 of Reviewer Y31x.)
>
> Weakness:
>
> 1. We appreciate the reference to Ma, Chen, and Fox 2015, and in our revised Section 3.1 we will both cite this paper and explain the distinction of our results from their Proposition.  Specifically, these authors assume a particular form of the key divergence-free field (G in our paper), and to do so they introduce a second “gamma’’ term to keep G divergence-free (see equation 3 therein). However, with this approach to the gamma term, it is not easy to relate the expressivity of a neural network’s “drift” (i.e., its dynamics, given by decay plus weights times activations) term with the score function (specifically, without knowledge of their Q matrix and its derivative).  In section 3.1 and appendix A, we take a novel and distinct approach, based on the Helmholtz-Hodge decomposition, that allows us to directly relate the score and required aspects of the network dynamics. Specifically, Section 3.1 shows that even if the neural dynamics are allowed to be irreversible (G is not 0), the function class of all dynamics still needs to have enough basis functions to approximate the score function part of the drift term that determines the stationary distribution.
>
> 2. We thank the reviewer for this critique and their important idea here. This has led us to an explicit new treatment of sampling speed (relaxation) to be included in our revision -- please see the new figure in the .pdf attached to our global reply.  We will also add text covering this important topic and the literature the reviewer notes in our revision. In more detail, following the reviewer's critique we have identified a simple way to increase sampling speed without additional optimization steps.  This is by letting the divergence-free (DF) field G in our paper be of the form $J\nabla p$ (treat Q as constant in Ma, Chen, and Fox 2015), where $J = -J^T$ can be any skew-symmetric matrix that is either learned through training or prescribed. Then the equation 14 in our paper would be $2\alpha (I + J)^{-1}(W_{out} \phi(\tilde{W}_{rec} x + I) - x) \approx \nabla\log p(x)$. The rest of the training procedure stays the same. Note that we do not need to alter the form of the dynamics in Equation 11. This improvement to the score matching loss gives a significant increase in sampling speed as we show in Figure S2.
>
> 3. As requested, we show the preliminary results of our reservoir sampler trained on the high-dimensional latent representations produced by a sparse coding model on MNIST (Fig. S4). Here, the latent space dimension is 3136 (four times overcomplete) and we used 30000 hidden neurons in the sampler.  We do note that this result is highly preliminary, and based on our work with the PCA projection we believe that it is possible that future work (on more powerful GPU systems) with larger reservoirs (though, of course, not yet approaching millions of units available in local circuits of the mammalian cortex) and/or other learning rules and hyperparameters are quite likely to succeed.
> In addition, we argue that MNIST is low-dimensional only if it undergoes a nonlinear transformation.  The first 300 PC components from MNIST just explain ~90% of the variance. Using a nonlinear autoencoder, we can indeed learn the distribution with a much smaller number (500) of reservoir neurons; see also Figure S3.
>
> 4. We thank the reviewer for this insightful comment: they are absolutely right that a tanh nonlinearity can lead to sampling of bimodal distributions.  However, another limitation does arise in this case: since tanh(x) and (x) are both odd functions, any distribution whose score function does not have a related symmetry property will not be able to be approximated (See Figure S1).
>
> Questions:
>
> - We thank the reviewer for pointing out the missing literature, it will be added in the camera-ready version.
>
> - The proof of theorem 3 is constructive in the sense that it constructs the weight of the RSN from a single-hidden-layer MLP.  We will add text to the revision to clarify this meaning.
>
> - Line 303-304 is referring to the fact that denoising score matching learns from gradually less noisy samples from a target distribution. Here we notice the striking similarity of this procedure to how the infant brain learns less noisy distribution of vision representations over time.
>
> - The main application of our work is to mechanistically model neural circuits that produce varied, non-Gaussian prior distributions that support Bayes-optimal behavior (reference 21, 26 in our paper)
>
> - With thanks, we will make the implied suggestions in the other careful reviewer questions as well.

---

> > ### Comment · Reviewer_pX3M · 2023-08-10
> >
> > I thank the authors for their thoughtful reply to my questions and those of the other reviewers. I appreciate their addition of tests relating to convergence time; I think this will enhance the final version. Given these additions, I will raise my score.
> >
> > A few small replies to individual points:
> > - Of course the choice of activation function will affect the choice of distributions that can be learned; I do not think the authors need to replace the existing ReLU figure. I think it is sufficient to note that for certain distributions the score can be realized exactly by sampler-only networks.
> > - I still do not follow the relationship with infant learning. This is, however, a minor point, so I do not think further elaboration is necessarily required.

---

> > > ### Author Response · Authors · 2023-08-11
> > > **Thank you**
> > >
> > > Thank you for your review and feedback. We agree with your suggestion to note the fact that sampler-only networks can sample exactly from some distributions. We will make sure to incorporate these changes in our revision.

---

### Official Review · Reviewer_kc8J · 2023-06-21

**Soundness:** 3 good
**Presentation:** 3 good
**Contribution:** 4 excellent
**Rating:** 7
**Confidence:** 2

**Summary:**

In this paper, the authors address the question of how the dynamics of recurrent neural networks (RNNs) can generate samples from probability distributions of interest. This is of interest to the neuroscience community, since it has been proposed that biological networks can use such sampling-based inference to represent and compute with probability distributions in order to account for their noisy surroundings. The authors prove that standard RNNs have limited expressivity when used directly for sampling-based inference.
They proceed to show that RNNs with separate (linear) readouts in contrast have the necessary expressivity to approximate any probability distribution through Langevin sampling. Finally, it is demonstrated empirically that such a 'reservoir' formulation allows RNNs to generate samples from a variety of complex distributions in practice, including multimodal, heavy-tailed, and high-dimensional distributions.

**Strengths:**

The paper contains several interesting theoretical results related to the topic of sampling-based inference using recurrent neural networks. This is an important open problem in computational neuroscience, where it remains unknown how the brain performs (approximate) probabilistic computations and represents the necessary probability distributions for such computations. By providing a firmer theoretical grounding for our understanding of sampling-based inference, this work has the potential to inspire both new computational and experiment studies of probabilistic computations in neural circuits. The paper is also generally well written and demonstrates impressive empirical results, although only on relatively 'toy' problems.

**Weaknesses:**

Much of the motivation and discussion of the paper is written with reference to the neuroscience literature on sampling-based inference, especially in sensory cortices. However, the empirical results all use artificial toy examples with no obvious relation to questions of interest to the neuroscience community. It would have been interesting to e.g. look at natural images or similar and delve a bit deeper into the dynamics learned by the network to make it more interesting to a neuroscience audience.

Many of the figure axis labels and legends are too small to be easily readable, and they would benefit from an increased font size.

**Questions:**

L60: The authors state that they derive a 'biologically plausible' learning rule, but the training procedure described in Eq 14 does not seem particularly biologically plausible, requiring both backprop and matrix inverses (as the authors also highlight themselves in L288-291). Perhaps the introduction could be reworded to make this initial statement more in tune with the subsequent methods/results?

L73: it would be useful to briefly state what these regularity conditions entail.

L114: Helmholtz typo

Figure 2 legend: "SO-SC and SO-FR is (_sic_) only able to fit the score function with piecewise linear function (_sic_) when using ReLU transfer function (_sic_)". Why the contrast between 'piecewise linear' in the legend and what looks like globally linear approximations to the score function in the figure?

Ref 5 & 37: formatting errors in author name ("GergHo Orban").

I am slightly confused by the authors' network dynamic equations (Eq 7 & Eq 8). When modeling neural firing rates, a positive (e.g. ReLU) nonlinearity is commonly used since firing rates are strictly positive. However, would the Brownian noise term added to the rates in Eq 8 not allow for negative neural firing rates?

I was also surprised by the synaptic current equations, since RNN dynamics are usually formulated in terms of either firing rates or membrane potentials. Potentials and currents are of course equivalent for ohmic units, but I was wondering if there was a reason that the authors decided to describe Eq 7 in terms of currents rather than membrane potentials?

The authors have discussed the case of a non-linear RNN with a linear readout and shown that it can approximate arbitrary probability distributions. Is the same true for non-linear RNNs with non-linear readouts? and if so, could the reservoir sampler network be considered a special case of a non-linear RNN where only a subset of units are used to represent samples from the target distribution, and the remainder are used for additional computational power/expressivity?

**Limitations:**

The authors have adequately addressed the limitations of their study.

---

> ### Author Rebuttal · Authors · 2023-08-09
>
> We appreciate the reviewer’s recognition of the significance of our work and the comments and suggestions. Below we address the reviewer’s concerns and questions.
>
> Weaknesses:
>
> - Regarding the application of our model to natural images:
> We thank the reviewer for bringing up this point. We agree that training our reservoir sampler on natural image patches would be more relevant for neuroscience applications. We plan to train the model on the CIFAR 10 dataset and evaluate the Fréchet inception distance
> (FID) of the model in the revised version.
>
> - We thank the reviewer for the accurate critique regarding our figure formatting and will fix the axis labels and legends in the revised version.
>
> Questions:
>
> - L60: The reviewer is absolutely correct that there is still considerable distance between our proposed algorithm and a biological implementation, and our revision will clearly note this.  Our meaning here was indeed more narrow than we now see we expressed, in that the learning rule “sidesteps the demands of backpropagation through time (BPTT)”.  We will carefully rephrase our usage of “biologically plausible” to make sure this qualification is clear throughout our revision.
>
> - L73: please see Cerrai, 2001 Hypothesis 2.1 & 2.2. Roughly this indicates that 1) the drift term b(x) does not increase too fast (slower than exponential rate) as x goes to infinity, 2) the drift term is locally Lipschitz, and 3) the diffusion coefficient is invertible as we assumed in our work.
>
> - L114: the typo will be fixed in the camera-ready version.
>
> - Figure 2 legend: We appreciate this astute point and the opportunity to clarify here; in addition, our revision will include a revised version of this figure that does not have the confusing linearity of the score function (please see Fig S1).  In reply, we first recall that the solutions are found by minimizing the denoising score-matching loss. Because we are using a distribution whose score function is symmetric with respect to the origin, it makes sense for the solution to have the same property.  With a piecewise linear activation function, SO-FR and SO-SC have the ability to “bend” the score function at a single point, but due to this symmetry they will not benefit from this in terms of the score matching loss, hence the seemingly “global linear approximation”. We believe that the new  Figure S1, which essentially conducts the same experiment using tanh nonlinearity instead of the ReLU function, presents the underlying situation much more clearly, and thank the reviewer for inspiring it.
>
> - Ref 5 & 37: will be fixed in the camera-ready version
>
> - Noise term can produce negative values within firing rate dynamics: We thank the reviewer for another on-target and careful point.  Yes indeed, this is true, and an imprecision (or flaw) that comes along with our basic SDE formulation of the rate-RNN on an unrestricted domain. In our revision, we will carefully note this.  We will also state that one quick remedy would be to add a high energy barrier at 0, i.e. add $C \cdot \operatorname{ReLU}(-x)$ to the score function where $C$ is large compared to the noise magnitude so that the dynamics would almost never output negative values. Since this only adds one more basis function, it would not affect our main argument.
>
> - Nomenclature of synaptic current dynamics: We absolutely agree that this terminology can be confusing!  After surveying a number of sources in the literature without identifying a clear consensus, we simply settled with the present choice as it follows the convention in the seminal text of Abbott & Dayan 2001 “Theoretical Neuroscience: Computational and Mathematical Modeling of Neural Systems” (cf. eq. 7.39).
>
> - What about non-linear readouts: If the nonlinear function following the linear layer is invertible (e.g. tanh), then the result still holds, and the density function would be scaled by the determinant of the jacobian. Otherwise, the result may not hold. For example, if we just apply ReLU to the sampler neurons, then we can only obtain positive samples. And yes, if we ignore the transients of the sampler neurons, then we can say that RSN is a special case of a non-linear RNN where only a subset of units are used to represent samples from the target distribution.

---

> > ### Comment · Reviewer_kc8J · 2023-08-10
> >
> > I appreciate the thorough response from the authors to myself and the other reviewers. The additional experiments with tanh nonlinearities and investigations of relaxation time will be particularly useful additions to the paper, and I look forward to seeing the results on the CIFAR 10 dataset when they are finished.
> >
> > I would also consider adding a sentence or two to the paper about the distinction between an RNN with a linear readout vs. an RNN where only a subset of neurons represent the target distribution, since the latter seems more intuitive from a neuroscience point of view.

---

> > > ### Author Response · Authors · 2023-08-11
> > > **Thank you**
> > >
> > > Thank you for your review and feedback. We agree with your suggestion for additional experiments on tanh nonlinearities, relaxation time, the CIFAR10 dataset, and more explanations of the distinction between RSN and sampling from part of a larger RNN.  We will make sure to incorporate these changes in our revision.

---

### Official Review · Reviewer_wwgi · 2023-07-03

**Soundness:** 3 good
**Presentation:** 3 good
**Contribution:** 3 good
**Rating:** 6
**Confidence:** 3

**Summary:**

- This study solves an outstanding problem involving arbitrary density sampling using stochastic neural networks.
- The authors propose a a Reservoir Sampling architecture, whereby an auxiliary recurrently-connected population facilitates the sampling from a non-trivial distribution.
- The work seems solid and elegant (although SDEs are not my domain of expertise), and the numerical experiments are convincing.

UPDATE: Sep 1, 2023. I have read the rebuttal, it addressed my Qs and I maintain my score.

**Strengths:**

- The math looks solid, although I did not have time to go through the supplement.
- There are numerical experiments on complex distributions (MNIST PCs), demonstrating that their procedure can operate on non-trivial densities.

**Weaknesses:**

- This work seems to address an _existence_ problem with arbitrary density sampling with RNNs, but how would one validate this kind of model in physiological experiments?
- In fact, I would appreciate more discussion on the state of affairs in the neurophysiology of density representation.

**Questions:**

- This auxiliary RNN population approach reminds me of a recent poster/paper at cosyne by Duong et al. ("Adaptive whitening in neural populations with gain-modulating interneurons"; ICML 2023 https://openreview.net/forum?id=cEWB5hABV5), where they show that if the auxiliary population is greater than a specific size, it can represent and whiten a multivariate gaussian densities exactly. In your framework, is there a relationship between the dimensionality of reservoir and the expressivity of the network in terms of what kinds of densities it can represent? Could this provide testable predictions for an experimentalist to search for?
- Suggestion: I did enjoy reading this manuscript, but there was more emphasis on technical parts than necessary to get your point across. If the text were reframed with less emphasis on the math, I think it would flow much nicer and would be much more approachable.

**Limitations:**

There was adequate discussion of limitations.

---

> ### Author Rebuttal · Authors · 2023-08-09
>
> We appreciate the reviewer’s critiques and suggestions. Our point-by-point replies follow, together with changes we will make to the revision in light of the reviewer’s points.
>
> Weaknesses:
>
> - Validation in physiology experiments: This is a great question and one which we now see we should have addressed fully in our submission; our revised manuscript will add to the discussion to improve the paper in this way.  In particular, we will describe the following general methodology: A first step in analyzing physiological data in our framework is to identify windows where the distribution of recorded neural activity is roughly stationary over time.  The next step is to train different network models using our framework to check if these models are capable of generating samples from the same stationary distribution as measured. Validation would involve not only asking whether this procedure succeeds, but evaluating the size of the “unobserved” reservoir network (i.e., number of hidden neurons) that is necessary, as well as whether different features of single-neuron physiology (modifications to the single-neuron and coupling terms that define the network) and priors on network connectivity lead to the better fitting of the experimental distribution.  We note that the outcome here is an answer to the question of what the conditions are on network size, together with the set of weights/other physiological parameters, that can reproduce the distribution of physiological data.  We do caution, however, that it will not be a complete answer, as failure to match these data could also arise from limitations in the underlying learning algorithm and/or optimization process.
>
> - Regarding the state of affairs in the neurophysiology of density representation:  We agree that providing more here would improve our paper, and in our revision, we will add text to the introduction and discussion to relay at least the following:  Strong evidence shows that both humans and animals use some form of uncertainty information to guide their behavior; moreover, multiple brain areas have been identified as neural correlates of this uncertainty [2]. Neurophysiology data and analyses that directly verify how neural activity represents probability densities remain in their early stages. However, neural sampling theories, to which our work contributes, have made empirical predictions of certain physiological properties such as noise correlations and Fano factors, and these have been verified in, e.g., [1]. We see our work as opening the door to an allied study of higher-order statistics that more fully describe complete probability distributions.
>
> Questions:
>
> - The higher the dimensionality is, the closer the stationary distribution of the sampler is to the target distribution because in this case, one has more basis functions at one’s disposal to match the score function. For testable predictions, a hallmark of whether enough neurons are considered is whether the distribution can indeed be captured; if not, our work suggests that might require a reservoir of neurons that supply input signals. Please also see further experimental implications under the response to your first “weakness” point above.
>
> - We are glad that the reviewer enjoyed reading our work, and appreciate the technical nature of much of our presentation as well.  For the revision, we will go through the paper top to bottom, and identify (1) at least several places where technical details can be moved to the supplement, and (2) clear topic sentences that can be added in less technical language that describe the import of a mathematical argument that follows, with the latter set off by a phrase such as “The technical details of this are as follows,” to accomplish the same objective.
>
> [1] Orbán G, Berkes P, Fiser J, Lengyel M. Neural Variability and Sampling-Based Probabilistic Representations in the Visual Cortex. Neuron. 2016 Oct 19;92(2):530-543. doi: 10.1016/j.neuron.2016.09.038. PMID: 27764674; PMCID: PMC5077700.
>
> [2] Rullán Buxó, Camille, and Cristina Savin. "A sampling-based circuit for optimal decision making." Advances in Neural Information Processing Systems 34 (2021): 14163-14175.

---

> > ### Comment · Reviewer_wwgi · 2023-08-10
> >
> > Thanks for your responses to my review.
> >
> > Because SDE's are not my expertise, there is a possibility that I'm missing technical details, so I will keep my score the same. But I will raise my confidence (2 -> 3) to reaffirm that I think the community would find this interesting.
> >
> > I do find the relation to neurophys and testable predictions a little weak, but that seems to just be where the field is at right now. I will reiterate that there is a larger audience that could appreciate this work, but some of the simple concepts are opaquely hidden behind the over-emphasis on technical presentation its present state. So, I hope the reviewers follow through on reformulating the presentation, including thinking about how to reach a broader neuroscience community and not just the neural probabilistic sampling community.

---

> > > ### Author Response · Authors · 2023-08-11
> > > **Thank you**
> > >
> > > Thank you for your review and feedback. We are in agreement with your suggestion to reformulate the manuscript for a broader neuroscience audience. We will make sure to incorporate these changes in our revision.

---

### Official Review · Reviewer_Y31x · 2023-07-06

**Soundness:** 4 excellent
**Presentation:** 3 good
**Contribution:** 3 good
**Rating:** 7
**Confidence:** 4

**Summary:**

The authors proposed a revervoir-sampler network whose firing rate dynamics can sample from an arbitrary probability distribution. They first established the relationship between the sampling power of the neural dynamics and the ability of the dynamics to approximate the score function. Then they showed that the synaptic current dynamics of the traditional sampler-only networks is only able to approximate score functions that are in a finite-dimensional function space. Importantly they proved that the  revervoir-sampler network can sample from an arbitrary probability distribution to arbitrary precision. Finally, they developped a biologically-plausible learning rule for the proposed model.

**Strengths:**

This paper is very clearly presented and the scientific question is quite interesting.

**Weaknesses:**

Comparison with previous models should be addressed in the experiment part.

**Questions:**

Q1: Why did the authors choose to use the reservoir network as the building block?

Q2: Following the first question, does the RSN model work because of the linear readout layer, which rescales the distributions? Can the authors explaine more about this？

Q3: Can the current model be extended to the multiplicative noise scenario which could be the feature of the poisson spike neurons, or (at the network level) be the feature of the E-I balanced spiking net.

Q4: How does the RSN model compare to the coupled attractor model which also implement Langevin sampling [1]?

[1]: Zhang, W. H., Lee, T. S., Doiron, B., & Wu, S. (2020). Distributed sampling-based Bayesian inference in coupled neural circuits. bioRxiv, 2020-07.


**Limitations:**

The authors addressed some limitations.

---

> ### Author Rebuttal · Authors · 2023-08-09
>
> We appreciate the reviewer for the thorough summary of our paper, as well as the comments and suggestions.
>
> Weakness:  We appreciate the importance of stronger comparisons with previous models, and in our revision, we will significantly expand our treatment of these in the main text. In the current version of the paper, the sampler-only architecture serves the role of previous models, and we discussed how previous works can be seen as special cases of either SO-SC or SO-FR model in Appendix D. In the revised version, we will include and expand the contents of Appendix D in the main text.
>
> Questions:
>
> Q1: Although the brain exhibits a hierarchical structure (e.g., the visual hierarchy), within each hierarchy neurons are recurrently connected, i.e. they may form a reservoir. This is our motivation for using the reservoir network as a basic building block of our model.
>
> Q2: Intuitively, the linear readout layer linearly combines different basis functions, and the reservoir is responsible for supplying a sufficient number of basis functions. Therefore the RSN model works because of 1) a large enough reservoir and 2) the linear readout layer.
>
> Q3: Thank you for this great question.  Yes, with a slight modification of the Fokker-Planck equation in the paper. If we take the diffusion coefficient to be $\sigma(x) = \sigma \sqrt{\operatorname{diag}(x)}$ and the diffusion matrix $\Sigma = \frac{1}{2} \sigma(x) \sigma(x)^T = \frac{1}{2} \sigma^2 \operatorname{diag}(x) $, it can be shown that the stationary distribution $p$ satisfies $\nabla \cdot(\frac{1}{2} \sigma^2 p + \Sigma \nabla p - p F) = 0$ where $F$ is the drift term of the neural dynamics. Again if we take the divergence field to be 0 and divide both sides by $p$, then the corresponding score matching problem would be $\nabla \log p \approx \Sigma^{-1} (F - \frac{1}{2} \sigma^2)$.
>
> Q4: Zhang et al. propose that coupled CANNs can implement Langevin sampling on the space of latent stimulus features. To see the connection between their work and ours, it is helpful to consider the functional form of the Langevin dynamics used by these authors (cf. eq. 4 in their paper). The score function is set directly by means of a formula to obtain the desired target distribution, and since only Gaussian distributions are considered, the stimulus feature needs only be linearly transformed (recall that the score function of Gaussian is linear). Our work, by contrast, (1) treats general target distributions, beyond the Gaussian case, (2) proposes a way to iteratively learn the score function by just looking at the samples from the target distribution, and (3) specifies conditions in which the score functions corresponding to general target distributions can and cannot be successfully learned.

---

### Official Review · Reviewer_sqZn · 2023-07-07

**Soundness:** 2 fair
**Presentation:** 3 good
**Contribution:** 3 good
**Rating:** 6
**Confidence:** 4

**Summary:**

The paper investigates architectural requirements for recurrent neural circuits to sample from complex distributions using diffusion models. It presents a model where traditional sampler-only networks are enhanced with additional firing-rate dynamics and a set of separate output units, called reservoir-sampler networks (RSNs). An efficient training procedure based on denoising score matching is proposed. Empirical experiments are presented to demonstrate the model's ability to sample from complex data distributions.

**Strengths:**


- The research addresses a significant issue in computational neuroscience and Bayesian learning – how neural dynamics can sample a complex distribution.
- Biological plausibility and mathematical computation efficiency are particularly commendable aspects of this work.
- The theoretical analysis is well constructed, providing strong support for the model design, including the choice of firing-rate dynamics and the reservoir-sampler network.
- The empirical experiments presented provide evidence of the effectiveness of the proposed method.

**Weaknesses:**

- The experiments conducted need improvements. Despite using score matching for model training, the quality of the generated images leaves room for improvement. The incorporation of Unet-like or transformer backbones might have enhanced the results.
- The paper lacks a clear, diagrammatic representation to explain the utilization of diffusion models in the design of recurrent neural circuits architecture.
- The results and conclusions seems to have limited impact to the general AI/ML areas, giving limited experimental results and comparison (e.g. precision and efficiency ) with existing diffusion probabilistic models is somewhat lacking.

**Questions:**

See weaknesses.

**Limitations:**

I am satisfied with the discussion of limitations.

---

> ### Author Rebuttal · Authors · 2023-08-09
>
> We thank the reviewer for the comments and suggestions. Below we address the reviewer’s concerns:
>
> Weakness:
> - Image quality is poor: It is an accurate point that the fidelity of the generated images is substantially lower than that would be realized by, for example, a U-net.  However, our goal here is to study probabilistic sampling in the class of RNNs that is broadly used as mechanistic and functional models in neurobiology.  Specifically, we ask: 1) whether the RNN that are considered in the current literature can sample from complex distributions, and 2) If not, what form of RNN is able to achieve this? While we believe our paper gives significant answers to both questions, ur, we fully acknowledge the subpar performance of using the RSN to generate complex images in contrast to U-nets.  We will acknowledge this explicitly in our revision and point the reader to the discovery of biological components that enable better score matching as an important area of further research.
>
> - We thank the reviewer for this important comment.   We wish to underscore that our approach, while partially inspired by the diffusion model, also holds differences. While the diffusion model can be conceptualized as a time-inhomogeneous SDE with a finite time horizon, our work delves into a time-homogeneous SDE (the neural dynamics) with an infinite time horizon. The entire training and sampling pipeline is diagrammatically described in Figure 1 and Figure 4a). Recognizing the need to make this more clear, in our revision, we will (1) more clearly refer to this diagram in the main text, and (2) add to the diagram a clear set of pseudocode.  We are confident that these two changes will both more clearly explain our methodology and illustrate to the readers the similarities and differences with the classical diffusion approach.
>
> - We appreciate that our emphasis is on implications for neuroscience, and as such we have limited implications for the cutting-edge AI/ML area in the short term. However, we do note that reviewers are starting to look at how sampling/diffusion can play a role in RL, robotics, and modeling human behavior [1-4]. Our paper explores the neural underpinnings of those applications and has the potential to inspire more AI/ML work in this regard.
>
> [1] Ajay, Anurag, Yilun Du, Abhi Gupta, Joshua Tenenbaum, Tommi Jaakkola, and Pulkit Agrawal. “Is Conditional Generative Modeling All You Need for Decision-Making?” arXiv, July 10, 2023. http://arxiv.org/abs/2211.15657.
>
> [2] Chi, Cheng, Siyuan Feng, Yilun Du, Zhenjia Xu, Eric Cousineau, Benjamin Burchfiel, and Shuran Song. “Diffusion Policy: Visuomotor Policy Learning via Action Diffusion.” arXiv, June 1, 2023. http://arxiv.org/abs/2303.04137.
>
> [3] Janner, Michael, Yilun Du, Joshua B. Tenenbaum, and Sergey Levine. “Planning with Diffusion for Flexible Behavior Synthesis.” arXiv, December 20, 2022. https://doi.org/10.48550/arXiv.2205.09991.
>
> [4] Pearce, Tim, Tabish Rashid, Anssi Kanervisto, Dave Bignell, Mingfei Sun, Raluca Georgescu, Sergio Valcarcel Macua, et al. “Imitating Human Behaviour with Diffusion Models.” arXiv, March 3, 2023. http://arxiv.org/abs/2301.10677.

---

> > ### Comment · Reviewer_sqZn · 2023-08-15
> > **Thanks**
> >
> > The response addresses most of my concerns, therefore I decide to keep my original ratings.

---

> > > ### Author Response · Authors · 2023-08-15
> > > **Thank you**
> > >
> > > We thank the reviewer for the review and feedback again.

---

### Author Rebuttal · Authors · 2023-08-09

We thank all reviewers for their expert review and insightful critiques, suggestions, and comments.  We have written a detailed, point by point reply to each reviewer's issue or concern in the replies to individual reviewers.

In the pdf attached to this general response, we illustrate the results requested by the reviewers.  Specifically, we show more results on the double peak experiment using tanh nonlinearity (Figure S1), accelerated sampling with irreversible dynamics (Figure S2), improved MNIST generation using autoencoder (Figure S3), and the preliminary result on the sparse coding setting (Figure S4).

---

### Decision · Program_Chairs · 2023-09-21

**Decision:**

Accept (poster)

**Comment:**

Recurrent neural circuits and algorithms are studied in the broad context of Bayesian sampling hypothesis of the brain. Authors demonstrated non-trivial distribution can be sampled. Reviewer-Author discussions have generated potential improvements and ideas and I suggest the authors put an extended amount of efforts in making the final version addressing the discussion on new comparisons and clarifications.